# Too Late to Recall: Explaining the Two-Hop Problem in Multimodal Knowledge Retrieval

**Constantin Venhoff**[1,4]    **Ashkan Khakzar**[1,4]    **Sonia Joseph**[2,3]    **Philip Torr**[1]    **Neel Nanda**

[1]University of Oxford    [2]McGill University    [3]Meta    [4]MATS

 cvenhoff/vlm-two-hop

## Abstract

Training vision language models (VLMs) aims to align visual representations from a vision encoder with the textual representations of a pretrained large language model (LLM). However, many VLMs exhibit reduced factual recall performance compared to their LLM backbones, raising the question of how effective multimodal fine-tuning is at extending existing mechanisms within the LLM to visual inputs. We argue that factual recall based on visual inputs requires VLMs to solve a two-hop problem: (1) forming entity representations from visual inputs, and (2) recalling associated factual knowledge based on these entity representations. By benchmarking 14 VLMs with various architectures (LLaVA, Native, Cross-Attention), sizes (7B-124B parameters), and training setups on factual recall tasks against their original LLM backbone models, we find that 11 of 14 models exhibit factual recall degradation. We select three models exhibiting high- and two models with low performance degradation, and use attribution patching, activation patching, and probing to show that degraded VLMs struggle to use the existing factual recall circuit of their LLM backbone, because they resolve the first hop too late in the computation. In contrast, high-performing VLMs resolve entity representations early enough to reuse the existing factual recall mechanism. Finally, we demonstrate two methods to recover performance: patching entity representations from the LLM backbone into the VLM, and prompting with chain-of-thought reasoning. Our results highlight that the speed of early entity resolution critically determines how effective VLMs are in using preexisting LLM mechanisms. More broadly, our work illustrates how mechanistic analysis can explain and unveil systematic failures in multimodal alignment.

## 1   Introduction

Vision–Language Models (VLMs) achieve strong multimodal task performance by integrating vision transformers (ViTs) with large language models (LLMs) via adapter mechanisms Liu et al. [2023], Lin et al. [2024], Steiner et al. [2024], Liu et al. [2024a], Cocchi et al. [2025], Touvron et al. [2023]. These adapters project visual representations from the ViT into the representation space of the backbone LLM, enabling text-based reasoning over visual inputs. However, how exactly preexisting mechanisms in the backbone LLM adapt to visual inputs, and the resulting implications or failure modes, remain severely understudied.

For example, previous studies have found that LLaVA-1.5-7B performs worse on factual recall tasks than its LLM backbone Cohen et al. [2024]. Explaining this requires understanding how visual information flows through VLMs and how factual recall mechanisms function in their LLM backbones. Prior work on factual recall mechanisms identified early-layer MLPs as a crucial component that

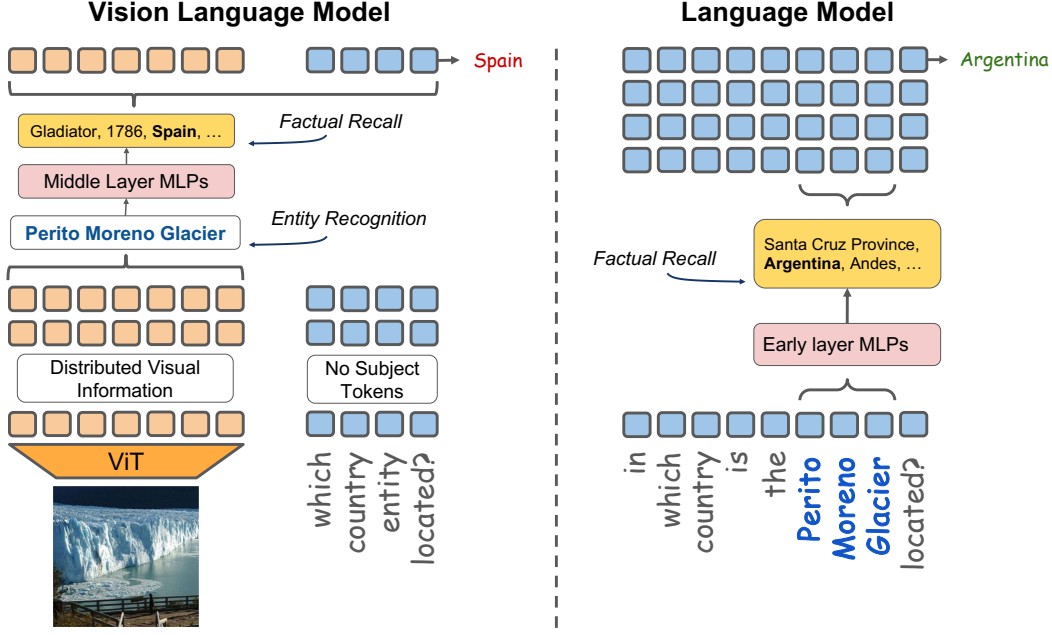

Figure 1: **Factual Recall in VLMs vs. LLMs – Illustration of the Two-Hop Problem.** This figure compares how a Vision Language Model (VLM, left) and a text-only Language Model (LLM, right) perform a factual recall task. The VLM receives an image of the Perito Moreno Glacier and a question: "In which country is the entity located?" The image is processed by a Vision Transformer (ViT), producing distributed visual embeddings that do not align with the LLM's pretrained token space. The entity ("Perito Moreno Glacier") is only recognized in the middle layers, bypassing the early-layer MLPs responsible for factual recall and resulting in an incorrect answer. In contrast, the LLM is given the full question, with the subject tokens "Perito Moreno Glacier" available. This enables early-layer MLPs to access factual knowledge ("Argentina") and produce the correct answer. The comparison highlights the core issue: VLMs must first infer the subject before retrieving facts, but because recognition occurs late, it cannot engage early factual recall mechanisms. This "two-hop" problem leads to degraded factual accuracy in VLMs, even when visual recognition succeeds.

reads in subject token representations and then produces subject-related factual representations Meng et al. [2022], Chughtai et al. [2024]. Separate work on VLM information flow, however, shows that visual projections are not aligned with the backbone LLM's token space and instead only gradually align with textual representations deeper in the LLM Venhoff et al. [2025], Wu et al. [2025], Masry et al. [2025].

Combining these observations, we hypothesize that VLMs fail to adapt to the preexisting factual recall circuit of their LLM backbone because visual representations in early layers are misaligned with the backbone LLM's token space and therefore "skip over" early-layer MLPs. Factual recall in VLMs can therefore be viewed as a two-hop problem: the VLM must first construct robust entity representations from visual inputs before it can access the LLM's preexisting factual knowledge.

To investigate this, we benchmark 14 VLMs across various architectures (LLaVA, Native, Cross-Attention), model sizes (7B–124B parameters), and training setups on factual recall tasks, comparing their performance to their original LLM backbone models. We find that 11 out of 14 models exhibit factual recall degradation. We then conduct a comparative analysis of well- and poorly-performing VLMs using attribution patching, activation patching, and linear probing. We find that VLMs with degraded performance rely on different sublayers for factual recall than their LLM backbones, while high-performing VLMs reuse the same components. In degraded VLMs, visual entity representations emerge too late in the backbone LLM's forward pass, bypassing early-layer MLPs of the factual recall mechanism, whereas well-aligned models exhibit early resolution of such representations. By patching early-layer MLP outputs from the backbone LLMs into degraded VLMs, we can recover factual recall performance, providing strong causal evidence for our hypothesis. Finally, we explore

Table 1: Factual-recall accuracy of VLMs vs. their text-only LLM backbones on 1000 questions.

| VLM Model | LLM Model | Architecture | LLM (%) | VLM (%) | Δ (%) |
|---|---|---|---|---|---|
| LLaVA-MORE-8B | Llama-3.1-8B-it | Adapter | 41 | 23 | ↓43.9 |
| LLaVA-NEXT-8B | Llama-3-8B-it | Adapter | 41 | 24 | ↓41.5 |
| LLaVA-1.5-7B | Llama-2-7B-chat | Adapter | 30 | 19 | ↓36.7 |
| LLaVA-1.5-13B | Llama-2-13B-chat | Adapter | 44 | 28 | ↓36.4 |
| Pixtral-Large-124B | Mistral-Large-2 | Adapter | 68 | 56 | ↓17.6 |
| Pixtral-12B | Mistral-NeMo | Adapter | 41 | 36 | ↓12.2 |
| Qwen2.5-VL-7B-it | Qwen2.5-7B-it | Adapter | 30 | 28 | ↓6.7 |
| Qwen2.5-VL-72B-it | Qwen2.5-72B-it | Adapter | 49 | 53 | ↑-8.2 |
| Llama-4-Maverick | — | Native | 75 | 71 | ↓5.3 |
| Gemini-2.0-Flash | — | Native | 66 | 63 | ↓4.5 |
| Gemma-3-27B-it | — | Native | 46 | 46 | 0.0 |
| Gemma-3-12B-it | — | Native | 40 | 40 | 0.0 |
| GPT-4o | — | Native | 77 | 73 | ↓5.2 |
| Llama-3.2-Vision-11B | Llama-3.1-8B-it | Cross-attn | 36 | 31 | ↓13.9 |

chain-of-thought prompting as a mitigation strategy, finding varying but promising improvements, particularly for larger VLMs.

Our findings highlight that successful multimodal alignment requires more than representational compatibility: it depends on integrating visual information into the functional circuits of the LLM backbone.

## 2 Benchmarking Factual Recall in Vision Language Models

To systematically evaluate factual-recall degradation in vision–language models (VLMs), we introduce a benchmark that directly compares each VLM with its original LLM backbone model. By feeding equivalent information to both systems, the benchmark isolates factual-recall ability from other confounding factors.

### 2.1 Benchmark Design

Our benchmark comprises 15000 multimodal factual-recall questions. We sample images from the Wikipedia-based Image–Text (WIT) dataset [Srinivasan et al., 2021] and use GPT-4.1 to generate entity-specific factual prompts (e.g. "Who invented the entity shown in the image?"). GPT-4.1 was chosen for its strong factual accuracy on the WildHallucinations benchmark [Zhao et al., 2024]. Appendix B details the data-generation pipeline, including entity verification and question construction.

During evaluation, a VLM is first asked to identify the main entity in each image. If the entity is misidentified, the sample is discarded to avoid conflating recognition errors with factual retrieval. We conduct a brief case study into rejected examples in Appendix G, to ensure there is no structural bias in the images discarded. For correctly identified entities, we compare the VLM's answer with that of its language-only backbone, ensuring a controlled, like-for-like comparison (for prompts see Appendix F).

We test a broad range of architectures, including *Adapter-based* VLMs, where a pretrained LLM is connected to a vision transformer (ViT) through lightweight adapters, *Native* VLMs, which are trained end-to-end with multimodal data from the beginning, and *Cross-Attention* VLMs where the LLM backbone uses cross-attention blocks to attend to representations of the ViT. Each VLM/backbone pair answers factual recall questions until 1000 valid samples are answered from the 15000-question pool.

### 2.2 Benchmark Results

Table 1 summarizes the results: 11 out of 14 VLMs exhibit lower factual-recall accuracy than their language-only backbones. Adapter-based- and Cross-Attention-models suffer the steepest drop.

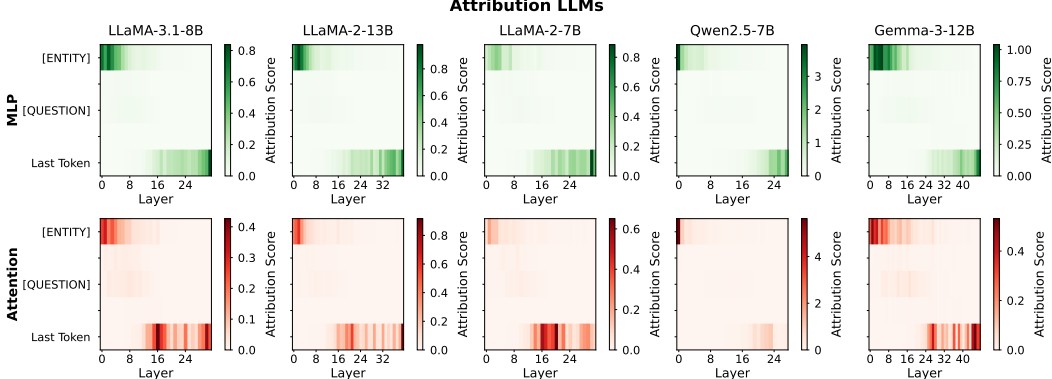

Figure 2: Attribution scores of each MLP and Attention sublayer for the original LLM backbone models. Higher values indicate higher causal relevance for factual recall.

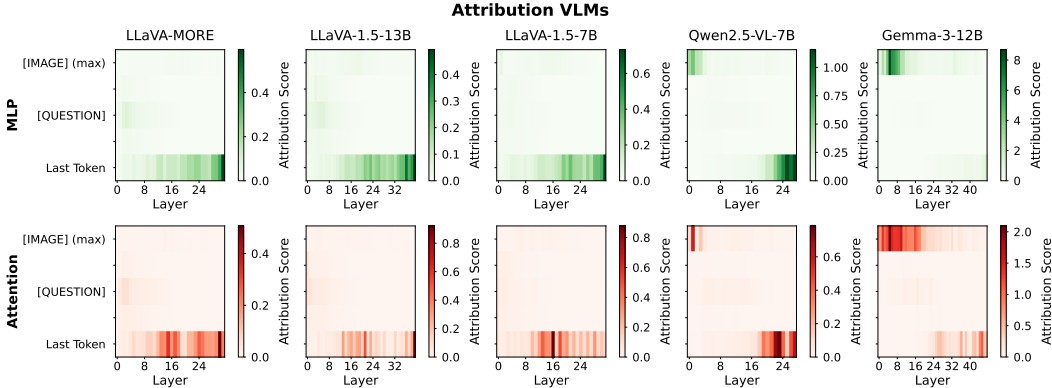

Figure 3: Attribution scores of each MLP and Attention sublayer for the VLM models. Higher values indicate higher causal relevance for factual recall.

Degradation persists even for very large models such as Pixtral-Large-124B. In general, Adapter-based models degrade more than Native VLMs, with the notable exception of Qwen2.5-VL. We hypothesize that Qwen2.5-VL's extensive multimodal fine-tuning on over 4 trillion tokens mitigates factual recall degradation, which may also explain why the Qwen2.5-VL-72B-it variant slightly *outperforms* its backbone.

In summary, almost all tested VLMs show pronounced factual-recall degradation, especially Adapter-based VLMs, while Native models such as Llama-4-Maverick and GPT-4o degrade less. Qwen2.5-VL's success suggests that massive multimodal fine-tuning can enable the VLM to utilize the factual knowledge of its backbone LLM.

## 2.3 Selecting Models for Comparative Analysis

In the following sections we describe several experiments to understand the drivers of factual recall degradation in VLMs. We select three Adapter-based models with very high degradation, LLaVA-MORE-8B [Cocchi et al., 2025], LLaVA-1.5-7B [Liu et al., 2023], and LLaVA-1.5-13B [Liu et al., 2023]. Additionally, we select two models with very low degradation, Gemma-3-12B-it [Team et al., 2025] to study Native models, and Qwen2.5-VL-7B-it [Bai et al., 2025] to understand the impact of massive multimodal fine-tuning on Adapter-based models.

# 3 Comparing Factual Recall Mechanisms

In this section we aim to find the sublayers the original LLM backbone model uses to perform factual recall, and compare them against the sublayers that the respective VLM model uses.

## 3.1 Attribution Patching

To determine which MLP- and Attention-sublayers in the LLM and VLM contribute most to factual recall, we employ an attribution patching methodology inspired by Nanda and Meng et al.. First we sample 100 correctly answered examples from the benchmark dataset for each VLM and their original LLM backbone model. Then we use the following steps to compute the attribution scores:

1. **Targeted corruption**: Let $\mathcal{S}$ be the token span necessary to identify the entity (LLM: entity text tokens; VLM: image tokens). Compute $\sigma_{\text{embed}}$ as the standard deviation of input embeddings over $\mathcal{S}$ across the examples of the attribution dataset. Perform a *corrupted run*, by injecting Gaussian noise in the embeddings across $\mathcal{S}$:

$$H_{\mathcal{S}}^{(0)} \leftarrow H_{\mathcal{S}}^{(0)} + \varepsilon, \quad \varepsilon \sim \mathcal{N}\big(0, \ \alpha\,\sigma_{\text{embed}}\big),$$

   where $\alpha$ is a noise multiplier. We ablate $\alpha$ for each model over the values $\{1, 2, \ldots, 9, 10\}$ and measure the corruption effect via the KL divergence between the clean predicted token distribution, and the corrupted predicted token distribution. More details on how we choose $\alpha$ can be found in Appendix D.

2. **Clean/corrupted passes and KL objective**: Run the model without the corruption (clean) and with the corruption (corrupted) to obtain next-token prediction logits $(z_{\text{clean}}, z_{\text{corr}})$ and define

$$L \ = \ D_{\text{KL}}\big(\text{softmax}(z_{\text{clean}}) \,\|\, \text{softmax}(z_{\text{corr}})\big).$$

3. **Per-layer branch activations and gradients**: For each layer $\ell$ and branch $b \in \{\text{mlp}, \text{attn}\}$, cache clean and corrupted outputs $H_{\ell,b}^{\text{clean}}$, $H_{\ell,b}^{\text{corr}}$ and backpropagate to obtain

$$G_{\ell,b} \ = \ \frac{\partial L}{\partial H_{\ell,b}^{\text{corr}}}.$$

4. **Grad×Delta attribution and aggregation**: For token $t$ and hidden dimension $d$,

$$a_{\ell,b}(t) \ = \ \Big| \sum_d G_{\ell,b}(t, d) \left( H_{\ell,b}^{\text{clean}}(t, d) - H_{\ell,b}^{\text{corr}}(t, d) \right) \Big|.$$

Figure 2 shows the attribution scores for the LLMs and Figure 3 shows the attribution scores for the VLMs averaged over the 100 correct factual recall examples from the benchmark dataset, aggregated across different token positions (entity/image, question, and last token positions).

## 3.2 Attribution Patching Results

We find the same two sites across all LLMs tested: the early-layer MLP- and Attention-sublayers across the entity tokens, and the middle-to-late layer MLP- and Attention-sublayers across the last token positions. This is consistent with findings from Meng et al. [2022]. For all LLaVA-style VLMs we find that there is only the mid-to-late layer site across the last token position active, indicating that the model is not able to use the same pathway as the LLM backbone. For Gemma-3-12B and Qwen2.5-VL-7B however, we find the same two sites, showing that the model uses a similar pathway, whether the entity is given in token space, or as an image.

# 4 Causal Analysis via Activation Patching

In Section 3, we found that for the three LLaVA models, the LLM backbone uses early-layer MLPs at the entity token positions to perform factual recall, while the LLaVA models that are trained on top of these LLMs are not able to do that. We hypothesize therefore, that these early-layer MLPs are a key breaking point in the factual recall mechanism of the LLaVA models. To test this, we patch MLP outputs from the LLM backbones of the three LLaVA models into their forward pass to see whether this yields relevant performance recovery, which would support that the LLaVA models are not able to utilize the factual recall circuit of their LLM backbone properly.

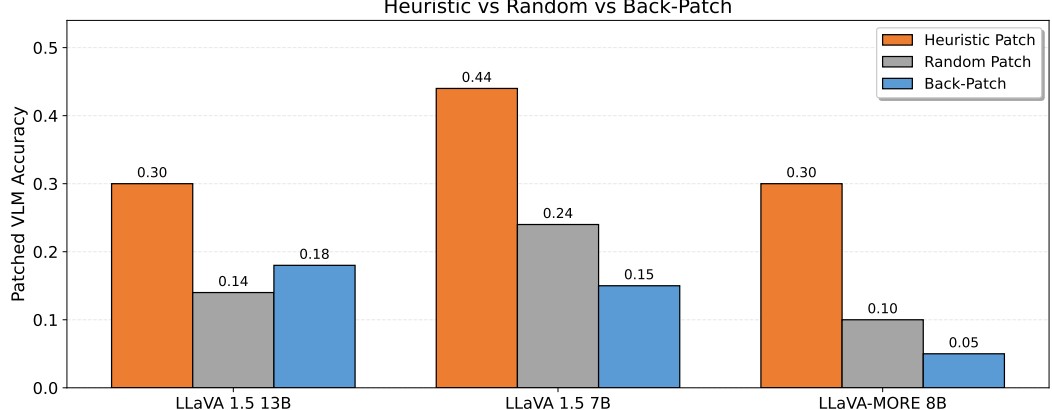

Figure 4: Factual recall performance recovery for LLaVA-1.5-7B, LLaVA-1.5-13B, and LLaVA-MORE models when MLP outputs across entity tokens from corresponding LLM backbones are patched into early VLM layers. We test only on examples where the VLM was originally wrong, so the y-axis directly shows the recovered performance gap. We compare our heuristic patching approach against a random baseline (randomly selecting patching positions), and back-patching.

## 4.1 Heuristic Patching

We propose *heuristic patching* to determine which token position in the VLM we patch the backbone LLM's MLP outputs (as the VLM doesn't have corresponding entity token positions). Heuristic patching uses 4 steps:

1. **Caching MLP outputs**: We define a set of source layers $\mathcal{L}$. For each layer $\ell \in \mathcal{L}$, cache the LLM's MLP outputs over the entity tokens.

2. **Define patching objective**: We define an objective, that quantifies whether patching the cached LLM MLP outputs into a given VLM token span, from a given start position $p$, increases the likelihood of the VLM to change its generated answer to the LLM answer. To do this, we obtain clean next-token predictions from both models. If the next-tokens predictions of the LLM and VLM differ, we use a logit-difference objective that increases the LLM-chosen token and decreases the VLM-chosen token. If they match, we use a KL objective over the LLM's top 10 tokens, renormalized. We treat both as a *signed* objective so that higher is better; we do not take absolute values.

3. **Position-wise attribution**: For every candidate token start position $p$ in the VLM, compute an attribution score by backpropagating the chosen objective to the VLM MLP outputs and taking a Grad$\times\Delta$, where $\Delta$ is the activation difference between LLM MLP activations and the current MLP activations across the VLM slice with start positions $p$. Average the scores across the slice token positions, keep the sign, and sum the scores across all layers in $\mathcal{L}$ for each token start position $p$ to obtain single scores $s(p)$.

4. **Selection and patching**: Choose the top start position with the highest $s(p)$. We patch *all* layers in $\mathcal{L}$ from the selected start token position during VLM generation and record whether the VLM now predicts the correct answer.

**Baselines.** The results of heuristic patching are hard to interpret in isolation, as 1) LLaVA models are fine-tuned end-to-end, hence the LLM backbone in the LLaVA model is not the same as the original LLM backbone model, and 2) there is no designated token position to patch into. Therefore it is unclear what accuracy to expect or consider *good*. Thus, we compare against (i) *Random patching*, which uses the same patching rule as heuristic patching, but chooses the start token positions $p$ uniformly at random, and (ii) *Back-patching*, a two-pass procedure both used in prior work to address two-hop problems in LLMs [Biran et al., 2024] and VLMs [Nikankin et al., 2025], that copies entire layer outputs across the image-tokens from a set of source layers $S$ to a set of (earlier) destination layers $D$. Concretely, we use:

- LLaVA-MORE (LLaMA-3.1-8B backbone): $S$=[6..16], $D$=[1..11].

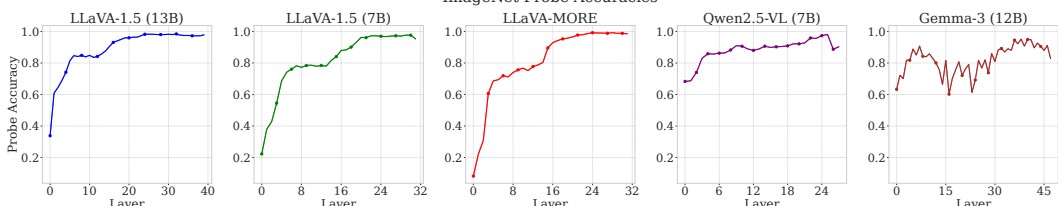

Figure 5: Accuracy of linear probes trained on residual-stream representations at each transformer layer of LLaVA-1.5-7B, LLaVA-1.5-13B, LLaVA-MORE, Gemma-3-12B, and Qwen2.5-VL-7B measured on ImageNet-100 entity prediction. The three LLaVA-style models exhibit a consistent pattern: probe accuracy remains poor in early layers and rises sharply between middle-to-late layers. Gemma-3-12B and Qwen2.5-VL-7B on the other hand show consistently high probe accuracies.

- LLaVA-1.5-7B (LLaMA-2-7B backbone): $S=[6..16]$, $D=[1..11]$.
- LLaVA-1.5-13B (LLaMA-2-13B backbone): $S=[27..37]$, $D=[13..23]$.

We select for each VLM/LLM pair a set of 100 factual recall questions from our dataset, which were incorrectly answered by the VLM and correctly answered by the corresponding LLM. Then we run heuristic patching with different layer choices (see Appendix E for details), and compare against the two baseline methods.

## 4.2 Patching Results

Results are shown in Figure 4. We find that on average heuristic patching recovers 35% of the factual recall accuracy difference between the VLM and its language-only counterpart, while back-patching only recovers 13%, and random patching recovers 16%. We therefore show that the early-layer MLP outputs used in the LLM's factual recall circuit are indeed a core missing piece in the degraded factual recall circuit used by the LLaVA-style VLMs.

## 5 Probing for the Emergence of Visual Entity Representations

Our previous experiments demonstrate that LLaVA-style VLMs show degraded factual recall performance, because they are not able to utilize the factual recall circuit of their LLM backbone correctly. The Native model Gemma-3-12B and the Adapter-based model Qwen2.5-VL-7B, trained on billions of image-text pair tokens, on the other hand, are seemingly able to use the same circuit, whether they are prompted with an image of an entity or the entity in text tokens.

We know already from prior work that the visual representations passed to the LLM backbone in LLaVA-style models are not aligned with the token space (reproduced in Appendix C). Therefore, in order to use the same circuit as the LLM backbone, VLMs need to produce robust entity representations in early layers of the LLM backbone. We hypothesize that the reason for the degraded performance of the LLaVA-style models is that entity representations emerge too late in the forward pass, compared to Gemma-3-12B and Qwen2.5-VL-7B which might be able to produce such representations much earlier due to native pretraining and massive multimodal fine-tuning respectively.

### 5.1 Linear Probing for Visual Entity Representations

To systematically assess where visual entity representations emerge within the LLM backbone of VLMs, we employ a probing methodology. Specifically, we train layer-wise linear classifiers (probes) on the residual stream outputs of each transformer layer to predict visual entities depicted in input images.

**Experimental Setup:** We use a subset of ImageNet-100 [Deng et al., 2009], restricted to 50 of the 100 classes, as our evaluation dataset, due to its controlled set of 50 classes and multiple images per entity, ensuring reliable estimation of representational capacity. We randomly select 2500 images and

Table 2: Factual-recall accuracy using *Chain-of-Thought (CoT) prompting*. Green arrows denote improvement compared to the base (non-CoT) runs.

| VLM Model | LLM Model | Architecture | LLM (CoT, %) | VLM (CoT, %) |
|-----------|-----------|--------------|--------------|--------------|
| LLaVA-MORE-8B | Llama-3.1-8B-it | Adapter | 46 ↑+5.0 | 28 ↑+5.0 |
| LLaVA-NEXT-8B | Llama-3-8B-it | Adapter | 45 ↑+4.0 | 30 ↑+6.0 |
| LLaVA-1.5-7B | Llama-2-7B-chat | Adapter | 35 ↑+5.0 | 18 ↓−1.0 |
| LLaVA-1.5-13B | Llama-2-13B-chat | Adapter | 45 ↑+1.0 | 37 ↑+9.0 |
| Pixtral-Large-124B | Mistral-Large-2 | Adapter | 72 ↑+4.0 | 71 ↑+15.0 |
| Pixtral-12B | Mistral-NeMo | Adapter | 45 ↑+4.0 | 48 ↑+12.0 |
| Qwen2.5-VL-7B-it | Qwen2.5-7B-it | Adapter | 35 ↑+5.0 | 43 ↑+15.0 |
| Qwen2.5-VL-72B-it | Qwen2.5-72B-it | Adapter | 52 ↑+3.0 | 56 ↑+3.0 |
| Llama-4-Maverick | — | Native | 79 ↑+4.0 | 77 ↑+6.0 |
| Gemini-2.0-Flash | — | Native | 70 ↑+4.0 | 70 ↑+7.0 |
| Gemma-3-27B-it | — | Native | 53 ↑+7.0 | 57 ↑+11.0 |
| Gemma-3-12B-it | — | Native | 41 ↑+1.0 | 46 ↑+6.0 |
| GPT-4o | — | Native | 79 ↑+2.0 | 79 ↑+6.0 |
| Llama-3.2-Vision-11B | Llama-3.1-8B-it | Cross-attn | 42 ↑+6.0 | 40 ↑+9.0 |

generate for each image a factual recall question, using a similar methodology as in Section 2. We provide GPT-4.1 with the entity class, and prompt for a concise factual recall question. We keep track of the last few generated questions and prompts to produce a different type of question, to ensure sufficient question diversity. At each transformer layer, we train an independent linear probe on the extracted and averaged residual-stream representations over the question token positions, to predict the correct entity label. We use a 20%/80% train-test split to evaluate the probes.

**Results and Analysis:** Figure 5 depicts the accuracy of the linear probes across layers for LLaVA-1.5-7B, LLaVA-1.5-13B, LLaVA-MORE, Gemma-3-12B, and Qwen2.5-VL-7B. The results reveal a clear trend for the LLaVA-style models: linear representations capable of reliably encoding visual entity information do not emerge until the middle-to-late layers. Prior to these middle-to-late layers, probe accuracy remains poor, indicating that early layers do not encode robust entity representations. Gemma-3-12B and Qwen2.5-VL-7B on the other hand show consistently high probe accuracies across all layers (with small variance), indicating that these models are capable of producing robust entity representations from visual inputs in the LLM backbone immediately.

These findings reveal the core challenge of the **two-hop problem** in multimodal knowledge retrieval: the model must first form a robust entity representation (first hop) before being able to access the latent knowledge of the LLM backbone encoded in early-layer MLPs (second hop). However, since entity representations only emerge in deeper layers in LLaVA-style models, they bypass the early-layer MLPs, causing factual recall degradation.

## 6   Reasoning as a Potential Mitigation

Previous experiments have shown that the LLaVA-style VLMs are fundamentally constrained in accessing the factual knowledge of their LLM backbone. The only mitigation seems to be either native pretraining, or massive multimodal fine-tuning, which both require orders of magnitude more compute and data than LLaVA-style models. Therefore, we experimented with another, simpler, mitigation strategy. The VLMs could use inference-time compute, to describe the visual entities relevant for factual recall in text space, and then answer the question. In this setup, the VLM might be able to access the factual recall knowledge of its LLM backbone again, as it produces token representations related to visual entities and therefore doesn't rely on pure visual representations anymore.

## 6.1 VLM Chain of Thought Prompting

**Experimental Setup:**  To test this, we alter the prompting template in the benchmark and include a chain of thought prompt (see Appendix F for prompt). Then we re-run the benchmark on the same data for each model and record the increase in factual recall accuracy.

**Results and Analysis:**  Table 2 shows the chain of thought benchmark accuracies for each model, alongside the increase in accuracy compared to the original accuracies without chain of thought. We find that for most models the chain of thought leads to a larger increase for the VLM compared to the LLM backbones, supporting our hypotheses. For Pixtral-12B and Pixtral-Large-124B we see that the performance gap is fully closed, while for the LLaVA-style models we see an increasing gain with the model size, with LLaVA-1.5-13B closing over half of its performance gap. Albeit having a smaller performance gap to begin with, we see that Gemini-2.0-Flash, GPT-4o, and LLama-4-Maverick close their respective performance gaps when using chain of thought prompting.

In summary, the effect of chain of thought prompting seems to be very model dependent (e.g. LLaVA-1.5-7B even *loses* a percentage of accuracy, as chain of thought prompting often leads to less structured responses), however models with substantial reasoning capabilities are able to close a significant fraction or even the full performance gap. We therefore think that reasoning is a promising approach for future research to improve factual recall performance of VLMs for which only limited data and compute resources are available.

# 7 Related Work

We review prior research on Vision Language Models (VLMs) and their handling of visual and textual representations. We first cover methods for aligning visual features with language embeddings and their effect on multimodal reasoning. We then summarize findings on factual recall in unimodal and multimodal models, highlighting evidence of recall degradation in VLMs.

## 7.1 Vision Language Models

VLMs integrate visual and textual information by mapping both modalities into a shared embedding space processed by a pretrained LLM Liu et al. [2023]. A key challenge is ensuring that visual features retain critical information (e.g., entities) when projected into the token space Masry et al. [2025]. The standard approach uses an image encoder and a projector (typically an MLP) to map visual data into the LLM's embedding space Liu et al. [2023, 2024b], Chen et al. [2024]. However, this mapping often fails to achieve meaningful alignment, limiting VLM performance on multimodal tasks Masry et al. [2025], Lin et al. [2024]. To improve this, several works propose alternatives that refine the mapping process or introduce cross-modal layers deeper in the network Alayrac et al. [2022], Yan et al. [2024], Li et al. [2024], Upadhyay et al. [2023]. While these methods enhance interaction between modalities, they add significant computational cost.

## 7.2 Multimodal Mechanistic Interpretability

Recent interpretability research has explored vision language models through various approaches. Chen et al. [2023] and Gandelsman et al. [2023] analyzed the vision encoder CLIP, identifying visual features Schwettmann et al. [2023] examined multimodal neurons in transformer models that were solely trained on text data. Jiang et al. [2024] investigated VLM responses to hallucinated versus real objects, and showed how to edit their internal representations to mitigate hallucinations. Work by Neo et al. [2024] investigated information flow within VLMs by projecting visual representations from intermediate layers onto language vocabulary, identifying gradual alignment with text vocabulary. Naghashyar et al. [2025] use sparse autoencoders to show how visual features emerge during multimodal fine-tuning in LLaVA.

## 7.3 Factual Recall in LLMs and VLMs

LLMs implement important mechanisms of factual recall in early-layer MLPs Chughtai et al. [2024], Meng et al. [2022], Geva et al. [2023]. This layer localization has been linked to their struggles with multi-hop reasoning, which requires resolving intermediate entities before answering the full query

Biran et al. [2024], Sakarvadia et al. [2023], Yang et al. [2024]. For example, answering "Who is the mother of the inventor of transistors?" requires first identifying the inventor. Biran et al. [2024] suggests early layers resolve these "bridge entities," leaving later layers with insufficient capacity for the follow-up reasoning Chughtai et al. [2024]. Cohen et al. [2024] reports that LLaVA-1.5-7B struggles with factual recall when processing images. Their patching experiments reveal that visual tokens stop influencing model behavior beyond middle layers, implying that visual processing finishes before later layers handle factual reasoning. They hypothesize that factual recall failures occur due to insufficient remaining layers after visual processing.

## 8    Discussion and Conclusion

Despite the growing capabilities of Vision Language Models (VLMs), their ability to reliably ground outputs in factual knowledge remains brittle. This is especially concerning in high-stakes applications, where factual hallucination carries real-world risks. In this work, we identify a structural failure mode in LLaVA-style VLMs: a systematic degradation of factual recall caused by insufficient early layer alignment of textual- and visual representations in the LLM backbone of the VLM.

Our findings show that such misalignment causes VLMs to use different sublayer-components than the original LLM backbone model to perform factual recall. This restricts them from accessing the full factual knowledge of their backbone LLM. We confirm this hypothesis using attribution patching, activation patching and probing experiments. We show that Native VLMs and adapter-based VLMs with massive multimodal fine-tuning do not exhibit the same degradation, as they are able to form consistent entity representations in early layers of the LLM backbone, regardless of the input modality.

These findings expose fundamental limitations in current LLaVA-style VLMs that are trained with limited data and compute resources. We conduct promising early experiments with chain of thought prompting to utilize inference-time compute to mitigate the problem, suggesting a potential avenue for future research. Additionally, future work should look into better, more data efficient, adapter mechanisms and multimodal alignment techniques to integrate visual representation earlier into the LLM's embedding space. While our study focuses on factual recall, future work should examine other tasks, comparing VLMs against their LLM backbone.

Overall, this study provides strong empirical evidence that representation misalignment limits VLMs to re-use the existing task circuits in their LLM backbone models. Further investigating and mitigating this issue is crucial for improving multimodal reasoning.

## Contributions

Constantin Venhoff conceived and led the project, designed the methodology, and conducted all experiments. Ashkan Khakzar and Sonia Joseph provided in-depth feedback throughout and contributed to the manuscript. Philip Torr and Neel Nanda advised on the research and provided high-level feedback.

## Acknowledgements

This work was carried out as part of the ML Alignment & Theory Scholars (MATS) program. We thank Josh Engels, Julian Minder, Clément Dumas, Iván Arcuschin, Nick Jiang and Sharan Maiya for helpful comments, discussions and feedback.

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

## A    Compute Resources

We used an NVIDIA A100 GPU with 80GB memory for all local experiments. Factual question generation for the benchmark was performed via API calls to GPT-4.1.

## B    Dataset Generation Pipeline

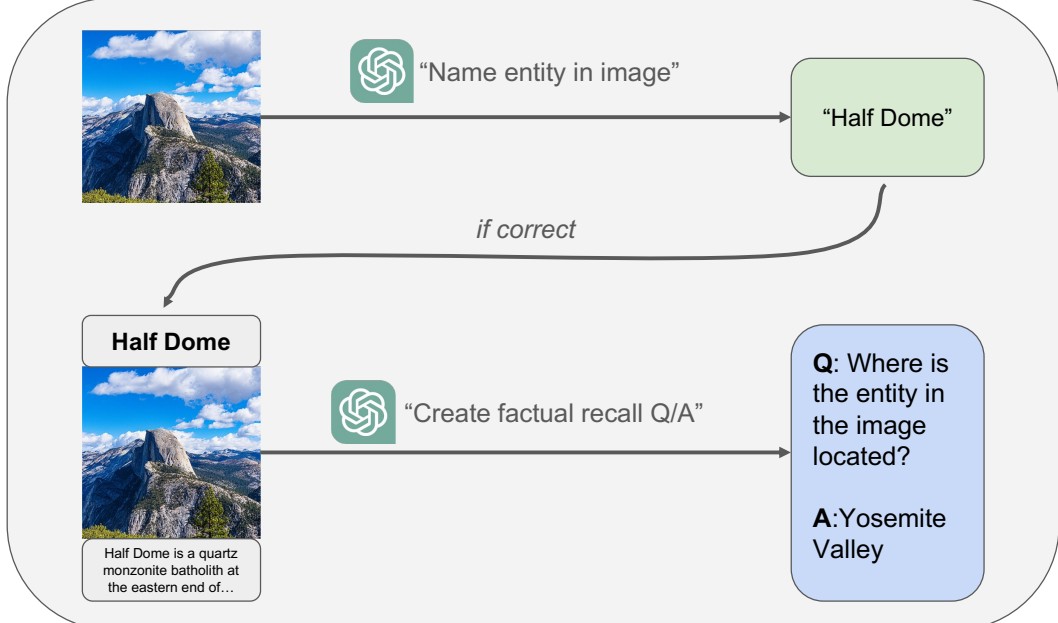

Figure 6:  Overview of the dataset construction pipeline for the multimodal factual recall benchmark.

We illustrate the pipeline to generate factual recall questions in Figure 6. The procedure has five core stages:

- **Image Sampling:** Images are randomly selected from WIT and paired with their ground-truth entity names from the page title;
- **Entity Detection:** GPT-4.1 is prompted to list the central identifiable entity in the image;
- **Entity Verification:** GPT-4.1 checks whether any detected entity matches (or paraphrases) the ground-truth entity name;
- **Question Generation:** Given the entity name and context from the Wikipedia page, a factual recall question about the entity is generated, without naming the entity explicitly;
- **Paraphrase Generation:** To minimize API calls during the benchmark run, we sample a list of common paraphrases for the entity name, enabling us to use simple string matching for entity guesses.
- **Save to Dataset:** The image, verified entity name (and paraphrases), and generated question are stored as a finalized benchmark sample.

## C    Token Space Misalignment

We reproduce the results of prior work [Masry et al., 2025], which shows that projector outputs of LLaVA-style VLMs are not aligned with the token space of their backbone LLMs, using LLaVA-1.5-7B, LLaVA-1.5-13B, and LLaVA-MORE. We extract visual projector outputs from 1,000 randomly selected ImageNet images. Each image produces 575 visual token activations, yielding a total of 575,000 activations per model. To assess alignment, we randomly sample 10,000 visual token embeddings per model and compute their cosine similarity with textual token embeddings from the LLM backbone.

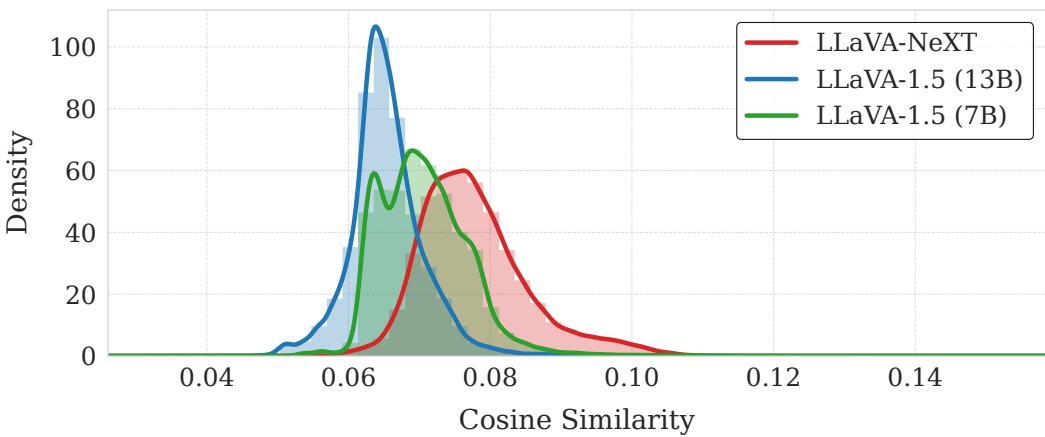

Figure 7: Distribution of absolute cosine similarity between visual projector outputs and text embeddings for LLaVA-1.5-7B, LLaVA-1.5-13B, and LLaVA-MORE. All models exhibit similar distributions, with means ranging from approximately 0.06 to 0.08. The distributions are characterized by their sharp, concentrated profiles with high density around their respective means. This concentrated distribution pattern strongly indicates that visual token embeddings consistently occupy subspaces nearly orthogonal to the pretrained text token embedding space across all evaluated models.

**Results and Analysis:** Figure 7 presents the distribution of cosine similarity scores between visual projector outputs and textual embeddings. The results indicate a pronounced misalignment across all evaluated models, with cosine similarity scores tightly concentrated near zero. This suggests that visual tokens predominantly occupy an embedding subspace orthogonal to pretrained textual representations.

These findings provide direct empirical support for our hypothesis that visual embeddings fail to integrate naturally into the LLM backbone's structured token space. As a result, early-layer factual recall mechanisms remain disengaged when processing visual inputs, reinforcing the idea that factual recall degradation in VLMs stems from fundamental adapter misalignment rather than insufficient computational depth alone.

## D    Attribution Patching Noise Multiplier

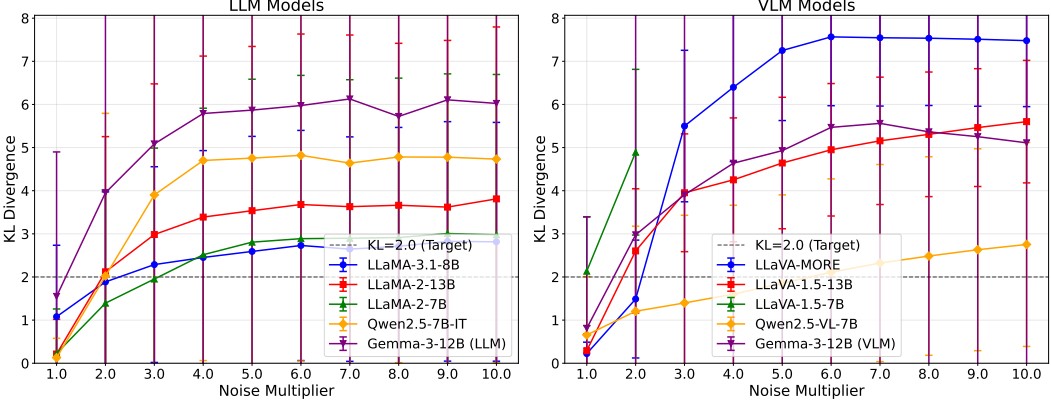

Figure 8: The x-axis shows different choices for the noise multiplier $\alpha$. The y-axis shows the average KL Divergence of the clean predicted token distribution and the corrupted predicted token distribution, using the indicated $\alpha$ for the corruption. (For LLaVA-1.5-7B we only show values for $\alpha \in \{1, 2\}$ as other values yielded NaN for the KL Divergence.)

The noise multiplier determines how strongly we corrupt the entity input embeddings (text tokens or image tokens). We aim for a choice that leads the model to predict a false answer, without completely corrupting its generated output. By experimenting with different values we find that a KL Divergence around 2 usually works well for the tested models. Therefore, we ablate $\alpha$ for each model and choose the value that causes a KL Divergence closest to 2. Results are shown in 8.

For LLaVA-1.5-7B we choose 1, for LLaVA-1.5-13B we choose 2, for LLaVA-More we choose 2, for Qwen2.5-7B-Instruct we choose 2, for Gemma-3-12B (LLM) we choose 1, for Qwen2.5-VL-7B-Instruct we choose 6, and for Gemma-3-12B (VLM) we choose 2.

# E   Layer Selection for Heuristic Patching

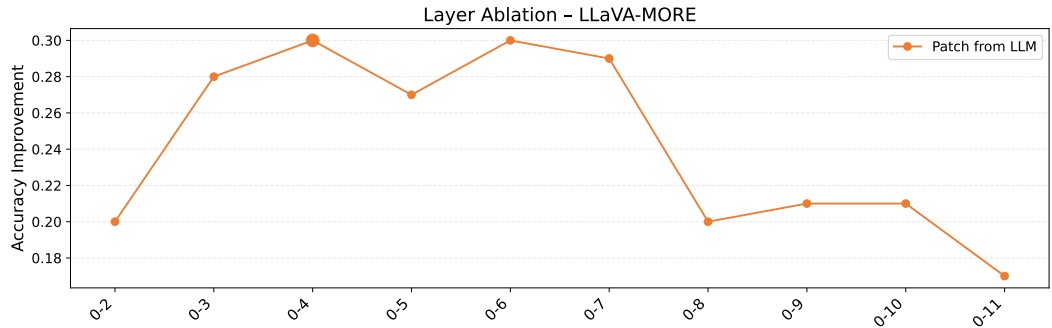

Figure 9: The x-axis shows the layer range used to cache and patch MLP activations from the original LLM backbone model into the according LLaVA-style VLM. The y-axis shows the recovered performance.

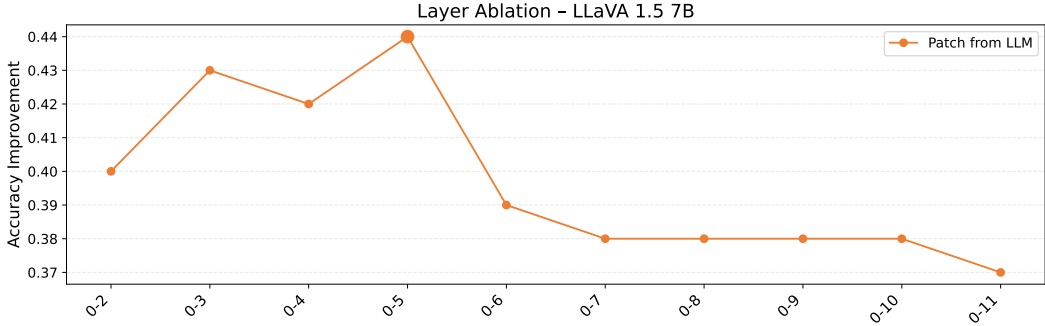

Figure 10: The x-axis shows the layer range used to cache and patch MLP activations from the original LLM backbone model into the according LLaVA-style VLM. The y-axis shows the recovered performance.

We run heuristic patching across several layer ranges, to capture the impact of the layer choice. For the final plot we use the best performing layer range. We see across all models a relatively consistent maximum, with a decrease in accuracy towards larger layer ranges, indicating that the early-layer MLPs are indeed causally most relevant for factual recall. Results for LLaVA-MORE are shown in 9, for LLaVA-1.5-7B in 10, and for LLaVA-1.5-13B in 11.

# F   Benchmark Prompt Templates

The placeholders {entity_name} and {question} are programmatically substituted at runtime.

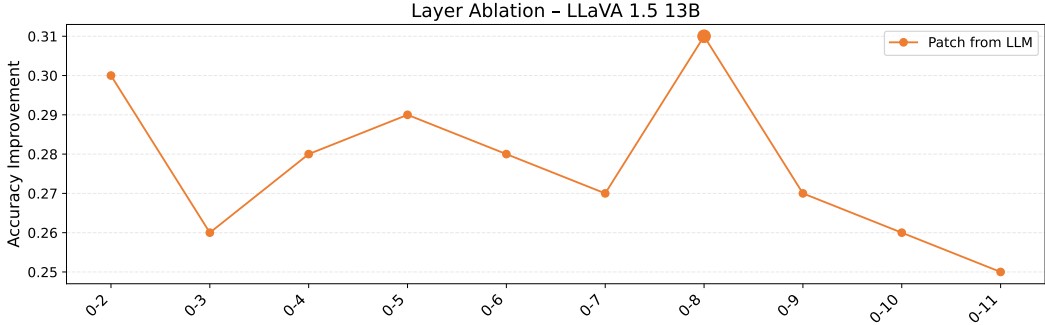

Figure 11: The x-axis shows the layer range used to cache and patch MLP activations from the original LLM backbone model into the according LLaVA-style VLM. The y-axis shows the recovered performance.

### F.1 Image entity recognition (VLM)

```
I provide you an image from wikipedia. Can you tell me what the specific
↪  name of the entity shown in the image is?
The specific name is the name the entity is commonly known by (name of
↪  person e.g. Julias Ceasar, name of location e.g. Eiffel Tower, name of
↪  building e.g. White House, etc.).

Required JSON fields:
- answer: A string containing the specific entity name

Example 1:
Input: Image of the Eiffel Tower in Paris
Output: {{
    "answer": "Eiffel Tower"
}}

Example 2:
Input: Image of a famous scientist
Output: {{
    "answer": "Albert Einstein"
}}

Answer in the following json format:
{{
    "answer": "<specific_entity_name>"
}}
```

### F.2 LLM factual recall (base)

```
Consider the following entity: {entity_name}.
Answer the following question: {question}

Required JSON fields:
- answer: A string containing the final answer in 1-5 words

Example 1:
Input: Entity: Eiffel Tower, Question: When was it completed?
Output: {{
    "answer": "1889"
}}

Example 2:
```

```
Input: Entity: Albert Einstein, Question: What theory is he most famous
↪  for?
Output: {{
    "answer": "Theory of Relativity"
}}

PROVIDE YOUR FINAL ANSWER IN 1-5 WORDS IN THE FOLLOWING FORMAT AND PROVIDE
↪  NO OTHER TEXT:
{{
    "answer": "<your answer>"
}}
```

## F.3   VLM factual recall (base)

```
I provide you with an image of an entity.
Answer the following question: {question}

Required JSON fields:
- answer: A string containing the final answer in 1-5 words

Example 1:
Input: Image of the Eiffel Tower, Question: When was it completed?
Output: {{
    "answer": "1889"
}}

Example 2:
Input: Image of Albert Einstein, Question: What theory is he most famous
↪  for?
Output: {{
    "answer": "Theory of Relativity"
}}

PROVIDE YOUR FINAL ANSWER IN 1-5 WORDS IN THE FOLLOWING FORMAT AND PROVIDE
↪  NO OTHER TEXT:
{{
    "answer": "<your answer>"
}}
```

## F.4   LLM factual recall (CoT)

```
Consider the following entity: {entity_name}.
Answer the following question: {question}
Reason through this question by:
1. Summarizing relevant background knowledge about the entity
2. Deriving the answer to the question

Required JSON fields:
- reasoning: A string containing 2-3 sentences explaining your thought
↪  process
- answer: A string containing the final answer in 1-5 words

Example 1:
Input: Entity: Eiffel Tower, Question: When was it completed?
Output: {{
    "reasoning": "The Eiffel Tower was built for the 1889 World's Fair in
    ↪  Paris. Construction began in 1887 and took about 2 years to
    ↪  complete.",
    "answer": "1889"
}}
```

```
Example 2:
Input: Entity: Albert Einstein, Question: What theory is he most famous
↪  for?
Output: {{
    "reasoning": "Einstein revolutionized physics with his theory of
    ↪  relativity. His most groundbreaking work was the theory of general
    ↪  relativity, which he published in 1915.",
    "answer": "Theory of Relativity"
}}

PROVIDE YOUR REASONING OF 2-3 SENTENCES AND FINAL ANSWER IN of 1-5 WORDS IN
↪  THE FOLLOWING FORMAT AND PROVIDE NO OTHER TEXT:
{{
    "reasoning": "<your reasoning>",
    "answer": "<your answer>"
}}
```

### F.5  VLM factual recall (CoT)

```
I provide you with an image of an entity.
Answer the following question: {question}
Reason through this question by:
1. providing a brief description of the image
2. Identify the entity the question refers to and summarize relevant
↪  background knowledge about the entity
3. Derive the answer to the question

Required JSON fields:
- reasoning: A string containing 3-4 sentences explaining your thought
↪  process
- answer: A string containing the final answer in 1-5 words

Example 1:
Input: Entity: Eiffel Tower, Question: When was it completed?
Output: {{
    "reasoning": "This is an image of the Eiffel Tower in Paris, a
    ↪  wrought-iron lattice tower. It was built for the 1889 World's Fair
    ↪  and took about 2 years to construct, with completion in 1889.",
    "answer": "1889"
}}

Example 2:
Input: Image of Albert Einstein, Question: What theory is he most famous
↪  for?
Output: {{
    "reasoning": "This is an image of Albert Einstein, a renowned physicist
    ↪  known for his revolutionary contributions to physics. His most
    ↪  groundbreaking work was the theory of general relativity, which he
    ↪  published in 1915 and fundamentally changed our understanding of
    ↪  space and time.",
    "answer": "Theory of Relativity"
}}

PROVIDE YOUR REASONING OF 3-4 SENTENCES AND FINAL ANSWER IN of 1-5 WORDS IN
↪  THE FOLLOWING FORMAT AND PROVIDE NO OTHER TEXT:
{{
    "reasoning": "<your reasoning>",
    "answer": "<your answer>"
}}
```

# G   Case Study into Excluded Images

We conduct a brief case study into the images for which entities are not correctly identified by VLMs. We check the entity recognition results of Pixtral-12B, and find that about 24% of examples are excluded due to Pixtral-12B not identifying the correct entity. A manual audit of 100 rejected cases reveals four overall failure modes:

1. **Paraphrase mismatch** – e.g., "FotoArtFestival" → "6 Foto Art Festival"
2. **Near-entity confusion** – e.g., "SS Monterey" → "RMS Mauretania"
3. **Imprecision** – e.g., "City Point, Wisconsin" → "Jackson County"
4. **Completely off-target** – rare; e.g., "Grand Palace Hotel" → "European Union"

We find that paraphrase mismatches and imprecision occur most often, likely because the VLM genuinely does not know the correct entity. Paraphrase mismatches in general are infrequent, which supports our approach of pre-generating a list of paraphrases to avoid an API call for every entity guess. Lastly, we find very few "completely off-target" cases. These seem to arise when the VLM misidentifies the central entity, e.g., the image above contains a European Union flag on the hotel, and the VLM assumes the target entity is the European Union rather than the hotel. No systematic bias toward any particular domain (e.g., geography, people, or protected classes) was observed. Images span landscapes, historical objects, banknotes, architecture, etc.; the error distribution looks random rather than skewed. Consequently, we believe these exclusions are unproblematic.

# NeurIPS Paper Checklist

1. **Claims**

   Question: Do the main claims made in the abstract and introduction accurately reflect the paper's contributions and scope?

   Answer: [Yes]

   Justification: The abstract and introduction accurately reflect the paper's contributions: (1) identifying factual recall degradation in VLMs using a new benchmark, (2) providing evidence of visual-text token misalignment, and (3) showing via probing and patching that visual entity representations emerge later than the early-layer factual recall mechanisms. These findings, presented in Sections 3–6, support the two-hop hypothesis through empirical evidence.

   Guidelines:

   - The answer NA means that the abstract and introduction do not include the claims made in the paper.
   - The abstract and/or introduction should clearly state the claims made, including the contributions made in the paper and important assumptions and limitations. A No or NA answer to this question will not be perceived well by the reviewers.
   - The claims made should match theoretical and experimental results, and reflect how much the results can be expected to generalize to other settings.
   - It is fine to include aspirational goals as motivation as long as it is clear that these goals are not attained by the paper.

2. **Limitations**

   Question: Does the paper discuss the limitations of the work performed by the authors?

   Answer: [Yes]

   Justification: The paper discusses several limitations in the Discussion and Conclusion section. It acknowledges that the study focuses primarily on LLaVA-style VLMs and suggests that future work should examine whether similar issues exist in other architectures. It also notes the limitation in scope regarding model scale and hints at the possibility that larger models may behave differently. Furthermore, it recognizes that alternative probing and intervention techniques may yield additional insights. While a separate "Limitations" section is not explicitly present, these points are conveyed within the discussion.

   Guidelines:

   - The answer NA means that the paper has no limitation while the answer No means that the paper has limitations, but those are not discussed in the paper.
   - The authors are encouraged to create a separate "Limitations" section in their paper.
   - The paper should point out any strong assumptions and how robust the results are to violations of these assumptions (e.g., independence assumptions, noiseless settings, model well-specification, asymptotic approximations only holding locally). The authors should reflect on how these assumptions might be violated in practice and what the implications would be.
   - The authors should reflect on the scope of the claims made, e.g., if the approach was only tested on a few datasets or with a few runs. In general, empirical results often depend on implicit assumptions, which should be articulated.
   - The authors should reflect on the factors that influence the performance of the approach. For example, a facial recognition algorithm may perform poorly when image resolution is low or images are taken in low lighting. Or a speech-to-text system might not be used reliably to provide closed captions for online lectures because it fails to handle technical jargon.
   - The authors should discuss the computational efficiency of the proposed algorithms and how they scale with dataset size.
   - If applicable, the authors should discuss possible limitations of their approach to address problems of privacy and fairness.

- While the authors might fear that complete honesty about limitations might be used by reviewers as grounds for rejection, a worse outcome might be that reviewers discover limitations that aren't acknowledged in the paper. The authors should use their best judgment and recognize that individual actions in favor of transparency play an important role in developing norms that preserve the integrity of the community. Reviewers will be specifically instructed to not penalize honesty concerning limitations.

3. **Theory assumptions and proofs**

   Question: For each theoretical result, does the paper provide the full set of assumptions and a complete (and correct) proof?

   Answer: [NA]

   Justification: The paper does not present any formal theoretical results, theorems, or proofs. Instead, it is an empirical and experimental study focused on benchmarking, latent analysis, attribution, probing, and patching experiments to support its claims. Therefore, the question is not applicable.

   Guidelines:

   - The answer NA means that the paper does not include theoretical results.
   - All the theorems, formulas, and proofs in the paper should be numbered and cross-referenced.
   - All assumptions should be clearly stated or referenced in the statement of any theorems.
   - The proofs can either appear in the main paper or the supplemental material, but if they appear in the supplemental material, the authors are encouraged to provide a short proof sketch to provide intuition.
   - Inversely, any informal proof provided in the core of the paper should be complemented by formal proofs provided in appendix or supplemental material.
   - Theorems and Lemmas that the proof relies upon should be properly referenced.

4. **Experimental result reproducibility**

   Question: Does the paper fully disclose all the information needed to reproduce the main experimental results of the paper to the extent that it affects the main claims and/or conclusions of the paper (regardless of whether the code and data are provided or not)?

   Answer: [Yes]

   Justification: The paper provides details to reproduce the main experimental results, including benchmark construction (2, with additional implementation details in Appendix B), patching methodology (4), attribution scoring (4), and probing setup (5). All evaluated datasets (ImageNet, CIFAR-100) are publicly available, and the question-generation procedure using GPT-4o is clearly described. The models used (LLaVA variants, GPT-4o, Claude Opus) are also publicly documented. Additionally, the code for all experiments and benchmark generation will be made publicly available to facilitate full reproducibility.

   Guidelines:

   - The answer NA means that the paper does not include experiments.
   - If the paper includes experiments, a No answer to this question will not be perceived well by the reviewers: Making the paper reproducible is important, regardless of whether the code and data are provided or not.
   - If the contribution is a dataset and/or model, the authors should describe the steps taken to make their results reproducible or verifiable.
   - Depending on the contribution, reproducibility can be accomplished in various ways. For example, if the contribution is a novel architecture, describing the architecture fully might suffice, or if the contribution is a specific model and empirical evaluation, it may be necessary to either make it possible for others to replicate the model with the same dataset, or provide access to the model. In general. releasing code and data is often one good way to accomplish this, but reproducibility can also be provided via detailed instructions for how to replicate the results, access to a hosted model (e.g., in the case of a large language model), releasing of a model checkpoint, or other means that are appropriate to the research performed.

- While NeurIPS does not require releasing code, the conference does require all submissions to provide some reasonable avenue for reproducibility, which may depend on the nature of the contribution. For example
  (a) If the contribution is primarily a new algorithm, the paper should make it clear how to reproduce that algorithm.
  (b) If the contribution is primarily a new model architecture, the paper should describe the architecture clearly and fully.
  (c) If the contribution is a new model (e.g., a large language model), then there should either be a way to access this model for reproducing the results or a way to reproduce the model (e.g., with an open-source dataset or instructions for how to construct the dataset).
  (d) We recognize that reproducibility may be tricky in some cases, in which case authors are welcome to describe the particular way they provide for reproducibility. In the case of closed-source models, it may be that access to the model is limited in some way (e.g., to registered users), but it should be possible for other researchers to have some path to reproducing or verifying the results.

5. **Open access to data and code**

   Question: Does the paper provide open access to the data and code, with sufficient instructions to faithfully reproduce the main experimental results, as described in supplemental material?

   Answer: [Yes]

   Justification: The paper provides open access to both the code and the benchmark dataset. These resources are included in the supplementary materials for reviewers and will also be made publicly available upon publication. The release includes scripts for data generation, experiment execution (e.g., probing and patching), and evaluation to faithfully reproduce the main experimental results.

   Guidelines:

   - The answer NA means that paper does not include experiments requiring code.
   - Please see the NeurIPS code and data submission guidelines (`https://nips.cc/public/guides/CodeSubmissionPolicy`) for more details.
   - While we encourage the release of code and data, we understand that this might not be possible, so "No" is an acceptable answer. Papers cannot be rejected simply for not including code, unless this is central to the contribution (e.g., for a new open-source benchmark).
   - The instructions should contain the exact command and environment needed to run to reproduce the results. See the NeurIPS code and data submission guidelines (`https://nips.cc/public/guides/CodeSubmissionPolicy`) for more details.
   - The authors should provide instructions on data access and preparation, including how to access the raw data, preprocessed data, intermediate data, and generated data, etc.
   - The authors should provide scripts to reproduce all experimental results for the new proposed method and baselines. If only a subset of experiments are reproducible, they should state which ones are omitted from the script and why.
   - At submission time, to preserve anonymity, the authors should release anonymized versions (if applicable).
   - Providing as much information as possible in supplemental material (appended to the paper) is recommended, but including URLs to data and code is permitted.

6. **Experimental setting/details**

   Question: Does the paper specify all the training and test details (e.g., data splits, hyperparameters, how they were chosen, type of optimizer, etc.) necessary to understand the results?

   Answer: [Yes]

   Justification: The paper provides all necessary details to understand and reproduce the experimental setup, including dataset sources, number of examples, model variants, and

evaluation methodologies. The probing setup is described in Section 5, patching methodology in Section 4, and benchmark construction in Section 2, with implementation details in Appendix B. As this is an evaluation and analysis paper—not focused on training new models—training-specific hyperparameters (e.g., optimizer, learning rate) are not applicable. All code, data generation scripts, and the benchmark dataset are included in the supplementary materials and will be made publicly available upon publication.

Guidelines:

- The answer NA means that the paper does not include experiments.
- The experimental setting should be presented in the core of the paper to a level of detail that is necessary to appreciate the results and make sense of them.
- The full details can be provided either with the code, in appendix, or as supplemental material.

7. **Experiment statistical significance**

Question: Does the paper report error bars suitably and correctly defined or other appropriate information about the statistical significance of the experiments?

Answer: [No]

Justification: The paper uses large, randomly sampled evaluation sets, and the observed differences—such as the VLM–LLM factual recall gap and patching improvements—are substantial and consistent across models (Sections 4, 2). Due to the clear empirical trends and large sample sizes, variance from different samplings was not expected to meaningfully affect conclusions. That said, we acknowledge that including statistical significance measures would strengthen the paper.

Guidelines:

- The answer NA means that the paper does not include experiments.
- The authors should answer "Yes" if the results are accompanied by error bars, confidence intervals, or statistical significance tests, at least for the experiments that support the main claims of the paper.
- The factors of variability that the error bars are capturing should be clearly stated (for example, train/test split, initialization, random drawing of some parameter, or overall run with given experimental conditions).
- The method for calculating the error bars should be explained (closed form formula, call to a library function, bootstrap, etc.)
- The assumptions made should be given (e.g., Normally distributed errors).
- It should be clear whether the error bar is the standard deviation or the standard error of the mean.
- It is OK to report 1-sigma error bars, but one should state it. The authors should preferably report a 2-sigma error bar than state that they have a 96% CI, if the hypothesis of Normality of errors is not verified.
- For asymmetric distributions, the authors should be careful not to show in tables or figures symmetric error bars that would yield results that are out of range (e.g. negative error rates).
- If error bars are reported in tables or plots, The authors should explain in the text how they were calculated and reference the corresponding figures or tables in the text.

8. **Experiments compute resources**

Question: For each experiment, does the paper provide sufficient information on the computer resources (type of compute workers, memory, time of execution) needed to reproduce the experiments?

Answer: [Yes]

Justification: The paper specifies that all experiments were conducted on an NVIDIA A100 GPU with 80GB memory. Since the work involves only inference (not training) on relatively small datasets (5K examples), the experiments are lightweight and not resource-intensive. Additionally, part of the benchmark generation involved API calls to GPT-4.1. Given the low computational demands, detailed runtime metrics were not included but the hardware setup is clearly documented for reproducibility.

Guidelines:

- The answer NA means that the paper does not include experiments.
- The paper should indicate the type of compute workers CPU or GPU, internal cluster, or cloud provider, including relevant memory and storage.
- The paper should provide the amount of compute required for each of the individual experimental runs as well as estimate the total compute.
- The paper should disclose whether the full research project required more compute than the experiments reported in the paper (e.g., preliminary or failed experiments that didn't make it into the paper).

9. **Code of ethics**

Question: Does the research conducted in the paper conform, in every respect, with the NeurIPS Code of Ethics https://neurips.cc/public/EthicsGuidelines?

Answer: [Yes]

Justification: The research conforms to the NeurIPS Code of Ethics. It involves standard datasets (ImageNet, CIFAR-100) and public vision-language models (e.g., LLaVA, GPT-4o) and does not involve human subjects, personally identifiable information, or ethically sensitive data.

Guidelines:

- The answer NA means that the authors have not reviewed the NeurIPS Code of Ethics.
- If the authors answer No, they should explain the special circumstances that require a deviation from the Code of Ethics.
- The authors should make sure to preserve anonymity (e.g., if there is a special consideration due to laws or regulations in their jurisdiction).

10. **Broader impacts**

Question: Does the paper discuss both potential positive societal impacts and negative societal impacts of the work performed?

Answer: [Yes]

Justification: The paper discusses broader societal implications in the Discussion and Conclusion section (see Section 8). It highlights the potential risks of factual recall failures in high-stakes settings, where misleading outputs from VLMs could have harmful consequences. At the same time, the work contributes positively by identifying architectural limitations that, if addressed, can improve the safety and reliability of future multimodal systems.

Guidelines:

- The answer NA means that there is no societal impact of the work performed.
- If the authors answer NA or No, they should explain why their work has no societal impact or why the paper does not address societal impact.
- Examples of negative societal impacts include potential malicious or unintended uses (e.g., disinformation, generating fake profiles, surveillance), fairness considerations (e.g., deployment of technologies that could make decisions that unfairly impact specific groups), privacy considerations, and security considerations.
- The conference expects that many papers will be foundational research and not tied to particular applications, let alone deployments. However, if there is a direct path to any negative applications, the authors should point it out. For example, it is legitimate to point out that an improvement in the quality of generative models could be used to generate deepfakes for disinformation. On the other hand, it is not needed to point out that a generic algorithm for optimizing neural networks could enable people to train models that generate Deepfakes faster.
- The authors should consider possible harms that could arise when the technology is being used as intended and functioning correctly, harms that could arise when the technology is being used as intended but gives incorrect results, and harms following from (intentional or unintentional) misuse of the technology.

- If there are negative societal impacts, the authors could also discuss possible mitigation strategies (e.g., gated release of models, providing defenses in addition to attacks, mechanisms for monitoring misuse, mechanisms to monitor how a system learns from feedback over time, improving the efficiency and accessibility of ML).

11. **Safeguards**

Question: Does the paper describe safeguards that have been put in place for responsible release of data or models that have a high risk for misuse (e.g., pretrained language models, image generators, or scraped datasets)?

Answer: [NA]

Justification: The paper does not release new models or datasets that pose high risks for misuse. It evaluates existing publicly available or proprietary models (e.g., LLaVA variants, GPT-4o, Claude Opus) and does not introduce new generative systems or web-scraped data. The released dataset is a benchmark constructed from a subset of ImageNet, which is a well-established and widely used dataset. As such, no special safeguards are necessary.

Guidelines:

- The answer NA means that the paper poses no such risks.
- Released models that have a high risk for misuse or dual-use should be released with necessary safeguards to allow for controlled use of the model, for example by requiring that users adhere to usage guidelines or restrictions to access the model or implementing safety filters.
- Datasets that have been scraped from the Internet could pose safety risks. The authors should describe how they avoided releasing unsafe images.
- We recognize that providing effective safeguards is challenging, and many papers do not require this, but we encourage authors to take this into account and make a best faith effort.

12. **Licenses for existing assets**

Question: Are the creators or original owners of assets (e.g., code, data, models), used in the paper, properly credited and are the license and terms of use explicitly mentioned and properly respected?

Answer: [Yes]

Justification: All external assets used in the paper—including models (e.g., LLaVA variants, GPT-4o, Claude Opus) and datasets (ImageNet, CIFAR-100)—are properly cited, and their licenses and terms of use are respected. ImageNet and CIFAR-100 are widely used public datasets with established licenses, and all referenced models are either open-source or accessed via official APIs under their respective terms.

Guidelines:

- The answer NA means that the paper does not use existing assets.
- The authors should cite the original paper that produced the code package or dataset.
- The authors should state which version of the asset is used and, if possible, include a URL.
- The name of the license (e.g., CC-BY 4.0) should be included for each asset.
- For scraped data from a particular source (e.g., website), the copyright and terms of service of that source should be provided.
- If assets are released, the license, copyright information, and terms of use in the package should be provided. For popular datasets, `paperswithcode.com/datasets` has curated licenses for some datasets. Their licensing guide can help determine the license of a dataset.
- For existing datasets that are re-packaged, both the original license and the license of the derived asset (if it has changed) should be provided.
- If this information is not available online, the authors are encouraged to reach out to the asset's creators.

13. **New assets**

Question: Are new assets introduced in the paper well documented and is the documentation provided alongside the assets?

Answer: [Yes]

Justification: The paper introduces a new benchmark for evaluating factual recall in VLMs, constructed from a subset of ImageNet with GPT-4.1-generated questions. The benchmark is well documented, with details provided in Section 2 and Appendix B. The dataset and associated scripts are included in the supplementary materials and will also be publicly released to ensure transparency and reproducibility.

Guidelines:

- The answer NA means that the paper does not release new assets.
- Researchers should communicate the details of the dataset/code/model as part of their submissions via structured templates. This includes details about training, license, limitations, etc.
- The paper should discuss whether and how consent was obtained from people whose asset is used.
- At submission time, remember to anonymize your assets (if applicable). You can either create an anonymized URL or include an anonymized zip file.

14. **Crowdsourcing and research with human subjects**

Question: For crowdsourcing experiments and research with human subjects, does the paper include the full text of instructions given to participants and screenshots, if applicable, as well as details about compensation (if any)?

Answer: [NA]

Justification: The paper does not involve any crowdsourcing or research with human subjects. All data used (ImageNet, CIFAR-100) is publicly available, and no new human annotations or interactions were collected for this work.

Guidelines:

- The answer NA means that the paper does not involve crowdsourcing nor research with human subjects.
- Including this information in the supplemental material is fine, but if the main contribution of the paper involves human subjects, then as much detail as possible should be included in the main paper.
- According to the NeurIPS Code of Ethics, workers involved in data collection, curation, or other labor should be paid at least the minimum wage in the country of the data collector.

15. **Institutional review board (IRB) approvals or equivalent for research with human subjects**

Question: Does the paper describe potential risks incurred by study participants, whether such risks were disclosed to the subjects, and whether Institutional Review Board (IRB) approvals (or an equivalent approval/review based on the requirements of your country or institution) were obtained?

Answer: [NA]

Justification: The paper does not involve research with human subjects and therefore does not require IRB approval. All experiments were conducted using publicly available datasets (ImageNet, CIFAR-100) and pre-existing models, with no human participation or data collection involved.

Guidelines:

- The answer NA means that the paper does not involve crowdsourcing nor research with human subjects.
- Depending on the country in which research is conducted, IRB approval (or equivalent) may be required for any human subjects research. If you obtained IRB approval, you should clearly state this in the paper.

- We recognize that the procedures for this may vary significantly between institutions and locations, and we expect authors to adhere to the NeurIPS Code of Ethics and the guidelines for their institution.
- For initial submissions, do not include any information that would break anonymity (if applicable), such as the institution conducting the review.

16. **Declaration of LLM usage**

Question: Does the paper describe the usage of LLMs if it is an important, original, or non-standard component of the core methods in this research? Note that if the LLM is used only for writing, editing, or formatting purposes and does not impact the core methodology, scientific rigorousness, or originality of the research, declaration is not required.

Answer: [Yes]

Justification: The paper uses GPT-4.1 to generate entity-specific factual questions as part of the benchmark construction process (see Section 2). This usage is a methodological component and is clearly described in the paper.

Guidelines:

- The answer NA means that the core method development in this research does not involve LLMs as any important, original, or non-standard components.
- Please refer to our LLM policy (`https://neurips.cc/Conferences/2025/LLM`) for what should or should not be described.

