# OpenReview forum: "Too Late to Recall: Explaining the Two-Hop Problem in Multimodal Knowledge Retrieval"
_NeurIPS.cc/2025/Conference — NeurIPS 2025 poster_

### Official Review · Reviewer_zisd · 2025-07-02

**Clarity:** 3
**Significance:** 3
**Originality:** 3
**Rating:** 4
**Confidence:** 4

**Summary:**

This paper investigates factual recall degradation in LLaVA-style Vision-Language Models (VLMs) compared to their underlying language models. The authors propose that this degradation stems from misalignment between visual projector outputs and pretrained textual embeddings, rather than insufficient computational depth as previously suggested. Through a systematic analysis involving cosine similarity measurements, a new benchmark, linear probing, and activation patching experiments, they demonstrate that visual entity representations emerge too late in the model's layers to engage early-layer factual recall mechanisms. The paper introduces the "two-hop problem" hypothesis: VLMs must first resolve visual inputs into entity representations before retrieving factual knowledge, but this resolution occurs after the critical early-layer recall mechanisms. Their patching experiments show ~40% recovery of the performance gap, providing causal evidence for their hypothesis.

**Questions:**

Have you tested or do you have hypotheses about whether this two-hop problem exists in non-LLaVA architectures like Flamingo that use different visual-linguistic integration mechanisms?

What factors contribute to the 60% of the performance gap that remains after patching? Have you explored combining your approach with other interventions?

In the ~35% of cases where VLMs successfully recall facts, what distinguishes these examples? Do certain types of entities or facts show better alignment timing?

Based on your findings, what specific architectural modifications would you recommend to address the timing misalignment? Would earlier cross-modal fusion help?

How does the two-hop problem manifest in larger VLMs? Does increased model capacity provide any natural mitigation?

**Ethical Concerns:**

["NO or VERY MINOR ethics concerns only"]

**Final Justification:**

The authors addressed my concerns. I would like to keep this score.

**Limitations:**

The authors adequately acknowledge that their study focuses on LLaVA-style models and suggest future work should examine other architectures. However, they could better discuss:

* The potential bias introduced by using GPT-4.1 for question generation
* The effectiveness and computational costs of their probing and patching approaches at scale
* Potential negative impacts if their insights are used to develop VLMs that hallucinate more convincingly

**Paper Formatting Concerns:**

No significant concerns.

**Quality:**

3

**Strengths And Weaknesses:**

Strengths:

The paper provides a compelling alternative explanation for VLM factual recall degradation, shifting focus from computational depth limitations to timing misalignment between visual processing and factual recall mechanisms.

The work combines multiple complementary approaches - cosine similarity analysis, benchmarking, probing, and causal intervention through patching - providing converging evidence for their hypothesis.

The factual recall benchmark with 1,000 examples provides a valuable resource for future research, with careful design to isolate factual recall from visual recognition errors.

The cross-model patching experiments provide compelling causal evidence, showing ~40% performance recovery when early-layer representations are restored.

The paper is well-written with effective visualizations, particularly Figure 1 which clearly illustrates the two-hop problem concept.

Weaknesses:

The paper only evaluates LLaVA-style architectures. It's unclear whether these findings generalize to other VLM architectures (e.g., cross-attn Flamingo, end2end chameleon, or more recent models).

The benchmark uses only ImageNet images and GPT-4.1 generated questions, which may not capture the full diversity of real-world factual queries about images. Also, how about object detection tasks?

---

> ### Author Rebuttal · Authors · 2025-07-31
>
> We sincerely thank **zisd** for the constructive suggestions.
>
> ---
> > **The benchmark uses only ImageNet images and GPT-4.1-generated questions, which may not capture the full diversity of real-world factual queries about images. Also, how about object-detection tasks?**
>
> We replace ImageNet with **WIT: Wikipedia-based Image–Text Dataset**, which comprises images of entities and their corresponding Wikipedia paragraphs. Instead of relying fully on GPT-4.1 to create a diverse and factually correct distribution of question–answer pairs, we now prompt it with both the image of an entity and its Wikipedia content. This ensures a diverse, realistic distribution of entities and factual grounding of questions in Wikipedia.
>
> We manually inspected hundreds of questions and found that GPT-4.1 adheres well to the given context; the questions consistently ask for factual knowledge mentioned in the Wikipedia paragraph.
>
> We agree that object detection could be another interesting instance of a two-hop problem. However, as most of **our** analysis is very specific to the factual-recall mechanism in the LLM backbone, adding more domains of two-hop questions is out of scope for this work—though an interesting avenue for future research.
>
> ---
> > **Have you tested or do you have hypotheses about whether this two-hop problem exists in non-LLaVA architectures like Flamingo that use different visual-linguistic integration mechanisms?**
>
> We ran the benchmark across a wide range of models: adapter-based (LLaVA, Pixtral, Qwen-VL), native (Llama-4, Gemma-3, Gemini-2.0) and a cross-attention model (Llama-3.2-Vision) covering sizes from 7 B to 124 B. Flamingo proved hard to elicit structured output from, so we used Llama-3.2-Vision as a cross-attention proxy.
>
> | **VLM Model**            | **LLM Model**      | **Arch.** | **LLM (%)** | **VLM (%)** | **Δ VLM→LLM** |
> |--------------------------|--------------------|-----------|-----------|-----------|--------------|
> | LLaVA-MORE-8B            | Llama-3.1-8B-it    | Adapter   | 41 | 23 | **–43.9** |
> | LLaVA-NEXT-8B            | Llama-3-8B-it      | Adapter   | 41 | 24 | **–41.5** |
> | LLaVA-1.5-7B             | Llama-2-7B-chat    | Adapter   | 30 | 19 | **–36.7** |
> | LLaVA-1.5-13B            | Llama-2-13B-chat   | Adapter   | 44 | 28 | **–36.4** |
> | Pixtral-Large-124B       | Mistral-Large-2    | Adapter   | 68 | 56 | **–17.6** |
> | Pixtral-12B              | Mistral-NeMo       | Adapter   | 41 | 36 | **–12.2** |
> | Qwen-2.5-VL-7B-it        | Qwen-2.5-7B-it     | Adapter   | 30 | 28 | **–6.7** |
> | Qwen-2.5-VL-72B-it       | Qwen-2.5-72B-it    | Adapter   | 49 | 53 | **+8.2** |
> | Llama-4-Maverick         | —                  | Native    | 75 | 71 | **–5.3** |
> | GPT-4o                   | —                  | Native    | 77 | 73 | **–5.2** |
> | Gemini-2.0-Flash         | —                  | Native    | 66 | 63 | **–4.5** |
> | Gemma-3-27B-it           | —                  | Native    | 46 | 46 | **0.0** |
> | Llama-3.2-Vision-11B     | Llama-3.1-8B-it    | X-attn    | 36 | 31 | **–13.9** |
>
> ### Findings
> - Factual-recall degradation appears in **11 of 13 models tested**.
> - The problem **extends to very large adapter-based models** (Pixtral-Large-124B, –17.6 %) and to the cross-attention model (–13.9 %).
> - Native models exhibit smaller drops (≈ 4–5 %), and Qwen-2.5-VL actually outperforms its backbone (+8.2 %), likely due to its > 4 T-token multimodal-alignment corpus.
>
> ### Summary
>
> 1. Factual-recall degradation is prevalent (11 / 13 models).
> 2. It affects large adapter-based and cross-attention models.
> 3. Native models, or adapter-based models trained on massive multimodal corpora, show less degradation.
>
> ---
> > **How does the two-hop problem manifest in larger VLMs? Does increased model capacity provide any natural mitigation?**
>
> As the previous section shows, very large models like Pixtral-Large-124B still exhibit significant factual-recall degradation, while smaller models like Qwen-VL-2.5-7B (with extensive alignment data) show less degradation.
>
> Therefore, we hypothesize that data mixture and the timing of modality fusion are crucial, even though the larger Qwen-2.5-VL-72B gains performance—implying that size is still beneficial given sufficient data.
>
> We also ran our probing experiment on Qwen-2.5-VL-7B and Gemma-3-11B to investigate why these models perform better than LLaVA.
>
> | Layer # (32 total) | 0 | 5 | 10 | 15 | 20 | 25 | 30 |
> |--------------------|----|----|----|----|----|----|----|
> | LLaVA-1.5-7B       | **0.18** | 0.40 | 0.81 | 0.96 | 0.98 | 1.00 | 1.00 |
>
> | Layer # (28 total) | 0 | 4 | 8 | 12 | 16 | 20 | 24 |
> |--------------------|----|----|----|----|----|----|----|
> | Qwen-2.5-VL-7B     | **0.78** | 0.91 | 0.96 | 0.99 | 1.00 | 1.00 | 1.00 |
>
> | Layer # (48 total) | 0 | 7 | 14 | 21 | 28 | 35 | 42 |
> |--------------------|----|----|----|----|----|----|----|
> | Gemma-3-11B        | **0.63** | 0.89 | 0.97 | 1.00 | 1.00 | 1.00 | 1.00 |
>
> Both Qwen-2.5-VL-7B and Gemma-3-11B form linear entity representations much earlier than LLaVA-1.5-7B, matching their stronger benchmark results. This supports that model size is **not** as important as data budget and/or timing of multimodal training, both of which help represent visual entities earlier and enable correct factual recall.
>
> ---
> > **Based on your findings, what specific architectural modifications would you recommend to address the timing misalignment? Would earlier cross-modal fusion help?**
>
> While our findings show that native training or large-scale multimodal training can help, this is often not feasible for smaller labs or open-source projects.
>
> Another idea is to utilize reasoning. If the VLM can articulate important entities robustly in LLM token space through reasoning before attempting to recall factual knowledge, it could fix the token-space misalignment. We tested reasoning as a lightweight fix across several models:
>
> | **VLM Model**        | **LLM (Base→CoT)** | **VLM (Base→CoT)** | **Δ Gap** |
> |----------------------|--------------------|--------------------|-----------|
> | LLaVA-MORE-8B        | 41 → +5 | 23 → +5 | –43.9 → –39.1 |
> | LLaVA-NEXT-8B        | 41 → +4 | 24 → +6 | –41.5 → –33.3 |
> | LLaVA-1.5-7B         | 30 → +5 | 19 → 0  | –36.7 → –45.7 |
> | LLaVA-1.5-13B        | 44 → 0  | 28 → +9 | –36.4 → –15.9 |
> | Pixtral-Large-124B   | 68 → +4 | 56 → +15| –17.6 → –1.3 |
> | Pixtral-12B          | 41 → +4 | 36 → +12| –12.2 → +6.6 |
> | Llama-3.2-Vision-11B | 36 → +6 | 31 → +9 | –13.9 → –4.7 |
>
> Reasoning recovers much of the gap for Pixtral-12B/124B, Llama-3.2-Vision-11B, and LLaVA-1.5-13B, suggesting an inexpensive mitigation. However, we find that LLaVA-1.5-7B reasons poorly: it often over-describes details and never names the entity. We believe that reasoning—perhaps reinforced with RL—is a promising future direction.
>
> ---
> > **What factors contribute to the 60 % of the performance gap that remains after patching? Have you explored combining your approach with other interventions?**
>
> We tried patching middle-layer attention heads at the final token positions. Although this recovers a few additional examples, it lags behind the MLP approach, and combining both does not improve the score.
>
> Our main hypothesis is that end-to-end visual instruction tuning slightly alters some attention heads and MLPs that were part of the factual-recall circuit, so full recovery is impossible without re-training.
>
> ---
> > **In the ~35 % of cases where VLMs successfully recall facts, what distinguishes these examples? Do certain types of entities or facts show better alignment timing?**
>
> In our last patching experiment we found that, for examples answered correctly by the VLM, linear probes can already recover the entity from early-layer visual representations.
>
> This is further supported by the models (Gemma-3, Qwen-2.5-VL) that show less degradation and also allow probing for linear entity representations in earlier layers—supporting this as the main difference between examples where factual recall works and where it fails. Which entities precisely enjoy superior early-layer representations likely depends on the data used for visual instruction tuning and multimodal pre-training.
>
> ---
>
> We hope these clarifications and additional results resolve the reviewer’s concerns and strengthen the paper. Thank you again for the thoughtful feedback.

---

> > ### Comment · Reviewer_zisd · 2025-08-08
> >
> > While the authors addressed surface concerns by testing more models and proposing reasoning solutions, their rebuttal lacks the mechanistic depth needed to elevate this work. They claim broad generalization by testing model variants rather than fundamentally different architectures, and crucially fail to explain why some models (Qwen-2.5-VL, Gemma-3) naturally avoid the two-hop problem. Their reasoning solution shows promise but lacks validation of whether it actually fixes the timing misalignment, and the 60% performance gap remaining after patching suggests their mechanistic understanding is incomplete. The response provides breadth over depth—more empirical coverage without the actionable insights needed for real impact.

---

> > > ### Author Response · Authors · 2025-08-08
> > >
> > > We appreciate that Reviewer zisd re-read our work and rebuttal, but we respectfully disagree with the claim that our response lacked "mechanistic depth" or that we only tested "model variants" rather than fundamentally different architectures.
> > >
> > > ## **On the breadth vs. depth concern**
> > > Our expanded benchmark explicitly includes three distinct classes covering different model architectures and training recipes, not just LLaVA variants:
> > > - **Adapter-based models** (LLaVA, Pixtral, Qwen-VL)
> > > - **Native end-to-end trained models** (Gemma-3, GPT-4o, Llama-4, Gemini-2.0)
> > > - **Cross-attention architectures** (Llama-3.2-Vision, a Flamingo-style integration)
> > >
> > > We emphasize that these are not simply parameter-scaled variants of the same design: they differ in fusion mechanism (adapter vs. cross-attention vs. fully native), training data regime (vision-only pretraining + alignment vs. joint multimodal pretraining), and in some cases optimization objectives.
> > >
> > > The persistence of factual recall degradation in adapter-based architectures, and its near-elimination only in native training or massive multimodal pretraining, directly supports our hypothesis that factual recall degradation stems from token-space misalignment.
> > >
> > > ## **On Qwen-2.5-VL and Gemma-3 avoiding the problem**
> > > Our rebuttal already provided mechanistic explanations for this: both models show *early-layer linear representations* of entities (via probing), unlike LLaVA, and correspondingly have minimal recall degradation. This is direct representational evidence for our claim: early entity representations allow the backbone's early-layer MLPs to recall the correct fact, while late emergence of linear representations skips over these essential MLPs.
> > >
> > > Across all models tested, the *only* consistent correlate of avoiding factual-recall degradation was the early emergence of linear entity representations in probing. Other factors, such as model size or instruction-tuning dataset, did not systematically explain the effect. This strongly supports our mechanistic explanation rather than alternative hypotheses.
> > >
> > > ## **On reasoning as a fix**
> > > We were clear that reasoning is a *practical* mitigation, which mitigates the issue by producing reasoning tokens that explicitly reference the visual entity. This reduces the two-hop problem back to the one-hop problem of LLM factual recall, which we have already covered in detail in the paper. It does not change the model's internal representations, since the model is not trained or altered in any other way. Therefore, claiming that we "lack validation of whether it actually fixes the timing misalignment" mischaracterizes reasoning as a mitigation strategy.
> > >
> > > ## **On the remaining 60% gap after patching**
> > > We agree that the patching performance gap needs context. As noted by other reviewers (e.g., ppcn), it is important to compare our patching results against baselines. We have already done so in our rebuttal, adding:
> > > - Random token patching → 4%
> > > - Back-patching (Nikankin et al. 2025) → 6%
> > > - Heuristic patching (ours) → 36%
> > >
> > > These baselines show that the recovery our method achieves is substantial and that early MLPs are the primary point of failure.
> > >
> > > Full recovery is not expected because the VLM has undergone end-to-end multimodal finetuning, parts of the original factual-recall circuit have been altered, which makes patching from the LLM not always well-defined.
> > >
> > > ## **Summary**
> > > Our work is not "breadth without depth." The breadth was added to *test* the mechanistic claim across different architectures; the depth comes from the convergence of evidence across multiple methodologies (probing, patching, cross-architectural analysis) that all support our mechanistic understanding. This convergent evidence from causal intervention (patching), probing experiments (adapter-based vs. native vs. large-scale pretraining), token-space alignment, and cross-architecture comparisons all tell the same story about early layer representation misalignment and factual recall.
> > >
> > > ---
> > >
> > > If the reviewer believes there are specific experiments missing, inconsistencies in how our evidence supports our hypothesis, or concrete reasons why our mechanistic account is insufficient, we would appreciate them being stated explicitly so we can address them directly.

---

### Official Review · Reviewer_ppcn · 2025-07-03

**Clarity:** 2
**Significance:** 3
**Originality:** 2
**Rating:** 4
**Confidence:** 4

**Summary:**

The paper investigates why LLaVA-style Vision-Language Models (VLMs) underperform at factual recall compared to their underlying LLM backbones. Rather than blaming insufficient compute after the visual encoder, the authors argue that misaligned visual token embeddings bypass the early-layer MLPs responsible for storing world knowledge. Through a series of linear‐probe analyses and “attribution patching” experiments, they show that (a) visual information only takes on a clean, linearly recoverable form in mid-to-late layers, and (b) if you inject the early MLP activations from the unimodal LLM into the VLM, you recover almost 40% of the factual recall gap. Finally, they release a benchmark that systematically compares VLM vs. LLM recall on 1,000 ImageNet-derived questions. Their main takeaway is that current adapter architectures delay entity formation until it’s too late to hook into the LLM’s fast recall pathways, creating a “two-hop” bottleneck.

**Questions:**

1. How uniform is the difficulty across GPT-4.1-generated questions? Could some entities consistently yield easier or harder questions, skewing VLM vs. LLM gaps?

2. Do findings hold for other adapter-based VLMs (e.g., Flamingo, InternVLM) or fusion architectures that interleave vision and language layers?

3. The “greedy matching” to align entity positions in the VLM seems critical. Can you quantify how often it picks the correct slot vs. noise spots, and does that affect recovery magnitude?

4. For the cases you exclude because the VLM misidentifies the entity, how often does that happen and does it correlate with certain image types?

5. Have you tried patching at a sliding window of layers (e.g., layers 1–3, 3–5, etc.) to pinpoint the minimal “hop” window needed for a meaningful recovery?

**Ethical Concerns:**

["NO or VERY MINOR ethics concerns only"]

**Final Justification:**

All my concerns have been resolved, but in the paper, the author said the code and benchmark data are included in the supplementary material, but I can't see them.

**Limitations:**

The authors acknowledge that their study is limited to LLaVA-style models and suggest future work should explore other architectures and scales. However, they do not address potential negative societal impacts of deploying misaligned VLMs (e.g., hallucinating harmful misinformation, misidentifying medical images).

**Quality:**

3

**Strengths And Weaknesses:**

Well-structured narrative: hypothesis → misalignment proof → patching → probing → success/failure comparison.

Focused solely on LLaVA variants; it’s unclear if patching tricks or adapter redesigns generalize to fusion-style VLMs.

Builds heavily on prior work without clearly delineating incremental vs. groundbreaking contributions

---

> ### Author Rebuttal · Authors · 2025-07-31
>
> We thank **ppcn** for the helpful suggestions and precise questions.
>
> ---
> > **How uniform is the difficulty across GPT-4.1-generated questions? Could some entities consistently yield easier or harder questions, skewing VLM vs. LLM gaps?**
>
> We replace ImageNet with **WIT: Wikipedia-based Image–Text Dataset**, which comprises images of entities and their corresponding Wikipedia paragraphs.  Instead of relying fully on GPT-4.1 to create a diverse and factually correct distribution of question–answer pairs, we now prompt it with both the image of an entity and its Wikipedia content. This ensures a diverse, realistic distribution of entities and factual grounding of questions in Wikipedia.
>
> We manually inspected hundreds of questions and found that GPT-4.1 adheres well to the given context; the questions consistently ask for factual knowledge mentioned in the Wikipedia paragraph.
>
> Because we test VLMs and LLMs on the exact same questions, providing the entity either visually or textually, the performance gap stems solely from the modality-specific performance difference.
>
> ---
> > **Do findings hold for other adapter-based VLMs (e.g., Flamingo, InternVLM) or fusion architectures that interleave vision and language layers?**
>
> We ran the benchmark across a wide range of models: adapter-based (LLaVA, Pixtral, Qwen-VL), native (Llama-4, Gemma-3, Gemini-2.0) and a cross-attention model (Llama-3.2-Vision) covering sizes from 7 B to 124 B. Flamingo proved hard to elicit structured output from, so we used Llama-3.2-Vision as a cross-attention proxy.
>
> | **VLM Model**            | **LLM Model**      | **Arch.** | **LLM (%)** | **VLM (%)** | **Δ VLM→LLM** |
> |--------------------------|--------------------|-----------|-----------|-----------|--------------|
> | LLaVA-MORE-8B            | Llama-3.1-8B-it    | Adapter   | 41 | 23 | **–43.9** |
> | LLaVA-NEXT-8B            | Llama-3-8B-it      | Adapter   | 41 | 24 | **–41.5** |
> | LLaVA-1.5-7B             | Llama-2-7B-chat    | Adapter   | 30 | 19 | **–36.7** |
> | LLaVA-1.5-13B            | Llama-2-13B-chat   | Adapter   | 44 | 28 | **–36.4** |
> | Pixtral-Large-124B       | Mistral-Large-2    | Adapter   | 68 | 56 | **–17.6** |
> | Pixtral-12B              | Mistral-NeMo       | Adapter   | 41 | 36 | **–12.2** |
> | Qwen-2.5-VL-7B-it        | Qwen-2.5-7B-it     | Adapter   | 30 | 28 | **–6.7** |
> | Qwen-2.5-VL-72B-it       | Qwen-2.5-72B-it    | Adapter   | 49 | 53 | **+8.2** |
> | Llama-4-Maverick         | —                  | Native    | 75 | 71 | **–5.3** |
> | GPT-4o                   | —                  | Native    | 77 | 73 | **–5.2** |
> | Gemini-2.0-Flash         | —                  | Native    | 66 | 63 | **–4.5** |
> | Gemma-3-27B-it           | —                  | Native    | 46 | 46 | **0.0** |
> | Llama-3.2-Vision-11B     | Llama-3.1-8B-it    | X-attn    | 36 | 31 | **–13.9** |
>
> ### Findings
> - Factual-recall degradation appears in **11 of 13 models tested**.
> - The problem **extends to very large adapter-based models** (Pixtral-Large-124B, –17.6 %) and to the cross-attention model (–13.9 %).
> - Native models exhibit smaller drops (≈ 4–5 %), and Qwen-2.5-VL actually outperforms its backbone (+8.2 %), likely due to its >4T-token multimodal alignment training corpus.
>
> ### Summary
>
> 1. Factual-recall degradation is prevalent (11 / 13 models).
> 2. It affects large adapter-based and cross-attention models.
> 3. Native models, or adapter-based models trained on massive multimodal corpora, show less degradation.
>
> ---
> > **The “greedy matching” to align entity positions in the VLM seems critical. Can you quantify how often it picks the correct slot vs. noise, and does that affect recovery magnitude?**
>
> There is no direct ground-truth slot, because we must patch LLM entity-token positions into VLM token positions. Since there is no entity token in the VLM template, we use a heuristic based on attribution to choose a slot.
>
> To quantify the heuristic as requested, we added two baselines:
>
> * **Random token patching** → **4%** average recovery
> * **Back-patching** (Nikankin et al. 2025) → **6%**
>
> The large gap between those results and our **36%** recovery rate, shows that our patching approach reliably selects a good slot.
>
> ---
> > **For cases excluded because the VLM misidentifies the entity, how often does that happen, and does it correlate with certain image types?**
>
> Using Pixtral-12B for entity recognition, about **24 %** of examples are excluded. A manual audit of 100 rejected cases reveals four failure modes:
>
> 1. **Paraphrase mismatch** – e.g., “FotoArtFestival” → “6 Foto Art Festival”
> 2. **Near-entity confusion** – e.g., “SS Monterey” → “RMS Mauretania”
> 3. **Imprecision** – e.g., “City Point, Wisconsin” → “Jackson County”
> 4. **Completely off-target** – rare; e.g., “Grand Palace Hotel” → “European Union”
>
> We find that paraphrase mismatches and imprecision occur most often, likely because the VLM genuinely does not know the correct entity. Paraphrase mismatches in general are infrequent, which supports our approach of pre-generating a list of paraphrases to avoid an API call for every entity guess. Lastly, we find very few “completely off-target” cases. These seem to arise when the VLM misidentifies the central entity, e.g. the image above contains a European Union flag on the hotel, and the VLM assumes the target entity is the European Union rather than the hotel.
>
> No systematic bias toward any particular domain (e.g., geography, people, or protected classes) was observed. Images span landscapes, historical objects, banknotes, architecture, etc.; the error distribution looks random rather than skewed. Consequently, we believe these exclusions are unproblematic for our gap analysis.
>
> ---
> > **Have you tried patching with a sliding window of layers (e.g., 1–3, 3–5) to pinpoint the minimal “hop” window needed?**
>
> We conducted an ablation study over the first ten layers (25 examples per window) on the three LLaVA models. Because the earliest layers are necessary and causally very relevant (attribution experiment), we ablated in an increasing window:
> (0, 1), (0, 1, 2), …, (0–9).
>
> | Layers window | 0-2 | 0-3 | 0-4 | 0-5 | 0-6 | 0-7 | 0-8 | 0-9 |
> |---------------|----|----|----|----|----|----|----|----|
> | **LLaVA-1.5-7B**   | 0.44 | 0.44 | 0.40 | **0.46** | 0.44 | 0.44 | 0.40 | 0.44 |
>
> | Layers window | 0-2 | 0-3 | 0-4 | 0-5 | 0-6 | 0-7 | 0-8 | 0-9 |
> |---------------|----|----|----|----|----|----|----|----|
> | **LLaVA-1.5-13B**  | 0.32 | 0.40 | 0.28 | 0.40 | 0.32 | 0.40 | **0.44** | 0.40 |
>
> | Layers window | 0-2 | 0-3 | 0-4 | 0-5 | 0-6 | 0-7 | 0-8 | 0-9 |
> |---------------|----|----|----|----|----|----|----|----|
> | **LLaVA-MORE-8B** | 0.40 | 0.28 | 0.40 | 0.40 | **0.44** | 0.40 | 0.32 | 0.28 |
>
> On 100 examples with the best window we obtain **0.37, 0.38, and 0.36** recovery for LLaVA-1.5-7B, LLaVA-1.5-13B, and LLaVA-MORE-8B, respectively.
>
>
> ---
> ## Some additional experiments
>
> ### Probing other architectures
> To improve generalization of our findings, we ran probing on Qwen-2.5-VL-7B (adapter-based, >4T alignment tokens) and Gemma-3-11B (native).
>
> | Layer # (32 total) | 0 | 5 | 10 | 15 | 20 | 25 | 30 |
> |--------------------|----|----|----|----|----|----|----|
> | LLaVA-1.5-7B       | **0.18** | 0.40 | 0.81 | 0.96 | 0.98 | 1.00 | 1.00 |
>
> | Layer # (28 total) | 0 | 4 | 8 | 12 | 16 | 20 | 24 |
> |--------------------|----|----|----|----|----|----|----|
> | Qwen-2.5-VL-7B     | **0.78** | 0.91 | 0.96 | 0.99 | 1.00 | 1.00 | 1.00 |
>
> | Layer # (48 total) | 0 | 7 | 14 | 21 | 28 | 35 | 42 |
> |--------------------|----|----|----|----|----|----|----|
> | Gemma-3-11B        | **0.63** | 0.89 | 0.97 | 1.00 | 1.00 | 1.00 | 1.00 |
>
> Both Qwen-2.5-VL-7B and Gemma-3-11B form linear entity representations much earlier than LLaVA-1.5-7B, matching their stronger benchmark results.
>
> ### Reasoning / chain-of-thought (CoT)
> Training a model natively or at Qwen scale is often infeasible, so we tested reasoning as a lightweight fix:
>
> | **VLM Model**        | **LLM (Base→CoT)** | **VLM (Base→CoT)** | **Δ Gap** |
> |----------------------|--------------------|--------------------|-----------|
> | LLaVA-MORE-8B        | 41 → +5 | 23 → +5 | –43.9 → –39.1 |
> | LLaVA-NEXT-8B        | 41 → +4 | 24 → +6 | –41.5 → –33.3 |
> | LLaVA-1.5-7B         | 30 → +5 | 19 → 0  | –36.7 → –45.7 |
> | LLaVA-1.5-13B        | 44 → 0  | 28 → +9 | –36.4 → –15.9 |
> | Pixtral-Large-124B   | 68 → +4 | 56 → +15| –17.6 → –1.3 |
> | Pixtral-12B          | 41 → +4 | 36 → +12| –12.2 → +6.6 |
> | Llama-3.2-Vision-11B | 36 → +6 | 31 → +9 | –13.9 → –4.7 |
>
> Reasoning recovers much of the gap for Pixtral-12B/124B, Llama-3.2-Vision-11B, and LLaVA-1.5-13B, suggesting an inexpensive mitigation.
>
> ---
>
> We hope these clarifications and additional results address all outstanding concerns and demonstrate the robustness of our findings. We are happy to answer any remaining questions. Thank you for your thoughtful review.

---

> > ### Comment · Reviewer_ppcn · 2025-08-07
> >
> > Thank the author for their rebuttal. All my concerns have been resolved, and I am willing to raise my score. I recommend that you integrate the new data and analysis into the final version of the paper or its appendix.

---

### Official Review · Reviewer_nkVQ · 2025-07-03

**Clarity:** 3
**Significance:** 3
**Originality:** 3
**Rating:** 4
**Confidence:** 4

**Summary:**

The paper investigates the factual recall degradation in LLaVA-style Vision-Language Models (VLMs) compared to their language backbone. It attributes this degradation to the architectural design of VLMs, where visual information is processed too late to effectively engage the early-layer factual recall mechanisms of the language model. The paper introduces the concept of a "two-hop" problem, provides empirical evidence through linear probing and patching experiments, and proposes a benchmark to evaluate factual recall accuracy and knowledge hallucination in VLMs. It also demonstrates that patching early-layer MLP outputs from the language model into the VLM can significantly improve factual recall performance.

**Questions:**

1. The authors should provide more details about the experiment setup.
2. The code and datasets are not included in the supplementary materials or publicly released.

**Ethical Concerns:**

["NO or VERY MINOR ethics concerns only"]

**Final Justification:**

I have read the author's rebuttal carefully, along with the other reviews, and most of my concerns have been resolved. The experiment on generalization is a good additional experiment. Given the paper's scope and comprehensive experiments, I remain mostly positive and will keep my scores.

**Limitations:**

yes

**Quality:**

3

**Strengths And Weaknesses:**

## **Summary Of Strengths:**

1. The paper presents a clear and novel explanation for the factual recall degradation in VLMs by highlighting the architectural limitations and the "two-hop" problem. This provides a fresh perspective on why VLMs underperform compared to their language backbones in factual recall tasks.
2. The paper support the hypothesis with a variety of empirical experiments, including linear probing, activation patching, and a newly introduced benchmark. These experiments help to validate the two-hop hypothesis and demonstrate the impact of visual-text misalignment in VLMs.
3. The introduction of fact recall benchmarks can better evaluate the accuracy of fact recall and knowledge illusion, which is of great significance for improving the reliability of VLMs.

## **Summary Of Weaknesses:**

1. While the paper provides empirical evidence for the two-hop problem, it lacks a more thorough comparison with other possible explanations for factual recall degradation in VLMs. This makes it difficult to determine if the two-hop problem is the sole or primary cause.
2. The models evaluated were small in size, and the authors should examine whether the behavior of larger models alleviates or exacerbates these problems.
3. The paper's focus is primarily on LLaVA-style VLMs, which limits the generalizability of the findings. It would be beneficial to explore whether similar issues exist in other VLM architectures.

---

> ### Author Rebuttal · Authors · 2025-07-31
>
> We truly thank **nkVQ** for the nice comments and helpful suggestions.
>
> We address all questions and concerns raised, specifically:
> 1. Generalization of findings
> 2. Considering alternative hypotheses
>
> ## 1. Generalization of findings
> We appreciate the suggestion to investigate the generalization of our results by including more diverse—and larger—models.
>
> To mitigate concerns about generality, we have:
> 1. Updated the benchmark dataset to ensure a more diverse, realistic, and factually grounded distribution.
> 2. Run the benchmark across a wide range of models, including adapter-based models (LLaVA, Pixtral, Qwen-VL), native models (Llama-4, Gemma-3, Gemini-2.0), and a cross-attention model (Llama-3.2-Vision) covering model sizes from 7B to 124B.
>
> Given inconsistencies in model classifications, we use:
> - **Adapter-based**: pretrained LLM + pretrained ViT, then finetuned together.
> - **Native**: the decoder is trained on text + vision from scratch (no standalone LLM).
> - **Cross-attention**: Flamingo-style cross-attention layers inject vision.
>
> ### Results
>
> | **VLM Model**            | **LLM Model**      | **Architecture** | **LLM (%)** | **VLM (%)** | **Δ VLM → LLM (%)** |
> |--------------------------|--------------------|------------------|-----------|-----------|---------|
> | LLaVA-MORE-8B            | Llama-3.1-8B-it    | Adapter          | 41 | 23 | **–43.9** |
> | LLaVA-NEXT-8B            | Llama-3-8B-it      | Adapter          | 41 | 24 | **–41.5** |
> | LLaVA-1.5-7B             | Llama-2-7B-chat    | Adapter          | 30 | 19 | **–36.7** |
> | LLaVA-1.5-13B            | Llama-2-13B-chat   | Adapter          | 44 | 28 | **–36.4** |
> | Pixtral-Large-124B       | Mistral-Large-2    | Adapter          | 68 | 56 | **–17.6** |
> | Pixtral-12B              | Mistral-NeMo       | Adapter          | 41 | 36 | **–12.2** |
> | Qwen-2.5-VL-7B-it        | Qwen-2.5-7B-it     | Adapter          | 30 | 28 | **–6.7** |
> | Qwen-2.5-VL-72B-it       | Qwen-2.5-72B-it    | Adapter          | 49 | 53 | **+8.2** |
> | Llama-4-Maverick         | —                  | Native           | 75 | 71 | **–5.3** |
> | GPT-4o                   | —                  | Native           | 77 | 73 | **–5.2** |
> | Gemini-2.0-Flash         | —                  | Native           | 66 | 63 | **–4.5** |
> | Gemma-3-27B-it           | —                  | Native           | 46 | 46 | **0.0** |
> | Llama-3.2-Vision-11B     | Llama-3.1-8B-it    | Cross-attention  | 36 | 31 | **–13.9** |
>
> ### Findings
>
> - Factual-recall degradation appears in **11 / 13 models tested**.
> - The issue **extends to very large adapter-based models** (Pixtral-Large-124B, –17.6 %) and to the cross-attention model (–13.9 %).
> - Native models drop only ≈ 4–5 %, and Qwen-2.5-VL actually beats its backbone (+8.2 %), likely due to its > 4 T-token multimodal alignment corpus.
>
> ### Summary
>
> 1. Factual-recall degradation is prevalent (11 / 13).
> 2. It affects large adapter-based and cross-attention models.
> 3. Native models—or adapter-based ones trained on massive multimodal corpora—show less degradation.
>
> Thus, unless one can afford native end-to-end training or huge multimodal datasets, understanding and mitigating factual-recall degradation remains critical.
>
> ## 2. Considering alternative hypotheses
>
> We thank nkVQ for suggesting adding experiments that challenge our main hypotheses and provide further context.
>
> ### 2.1  Late-layer representations instead of early-layer MLPs
> Our attribution experiment focuses solely on early-layer MLPs. However, prior work on two-hop queries in LLMs (Biran et al. 2024, https://arxiv.org/abs/2406.12775) and modality-specific mechanisms in VLMs (Nikankin et al. 2025, https://arxiv.org/html/2506.09047v1) shows that “back-patching” later-layer representations into earlier layers can also help recover performance.
> We implemented the exact back-patching setup of Nikankin et al. as a baseline and find that it closes only **6–8 %** of the performance gap, compared with **35–40 %** from our early-MLP patching.
> This further supports our hypothesis that early-layer MLPs are the sub-module that breaks in multimodal factual recall due to token-space misalignment, in contrast to the alternative explanation that VLMs could perform correct factual recall if they had only access to their later-layer representations.
>
> ### 2.2  Additional probing experiments
> Native VLMs and adapter-based VLMs with large-scale alignment show far less degradation, so we can use them as counterfactuals in the probing experiment. If our early-layer claim is right, probing should reveal that linear entity representations occur earlier in those models compared with the LLaVA-style VLMs.
>
> | Layer # (32 total) | 0 | 5 | 10 | 15 | 20 | 25 | 30 |
> |--------------------|----|----|----|----|----|----|----|
> | LLaVA-1.5-7B       | **0.18** | 0.40 | 0.81 | 0.96 | 0.98 | 1.00 | 1.00 |
>
> | Layer # (28 total) | 0 | 4 | 8 | 12 | 16 | 20 | 24 |
> |--------------------|----|----|----|----|----|----|----|
> | Qwen-2.5-VL-7B     | **0.78** | 0.91 | 0.96 | 0.99 | 1.00 | 1.00 | 1.00 |
>
> | Layer # (48 total) | 0 | 7 | 14 | 21 | 28 | 35 | 42 |
> |--------------------|----|----|----|----|----|----|----|
> | Gemma-3-11B        | **0.63** | 0.89 | 0.97 | 1.00 | 1.00 | 1.00 | 1.00 |
>
> Indeed, Qwen-2.5-VL-7B and Gemma-3-11B form linear entity representations far earlier than LLaVA-1.5-7B, matching their stronger benchmark scores and supporting our hypothesis.
>
> ### Summary
> We added the back-patching baseline from related work and a counterfactual probing study. These additions contextualize and support our claims and show that alternative hypotheses are not supported by the empirical results.
>
> ## Additional: reasoning / chain-of-thought (CoT)
> It is often not feasible to train a model natively or run large-scale alignment training like Qwen-2.5-VL. Therefore, we tested an alternative: reasoning. We reran the benchmark for models most affected by factual-recall degradation, adding a chain-of-thought prompt that lets the model reason about the image and question for a couple of sentences.
>
> | **VLM Model**           | **LLM (Base → CoT)** | **VLM (Base → CoT)** | **Δ Gap (%)** |
> |-------------------------|----------------------|----------------------|--------------|
> | LLaVA-MORE-8B           | 41 → +5              | 23 → +5              | –43.9 → –39.1 |
> | LLaVA-NEXT-8B           | 41 → +4              | 24 → +6              | –41.5 → –33.3 |
> | LLaVA-1.5-7B            | 30 → +5              | 19 →  0              | –36.7 → –45.7 |
> | LLaVA-1.5-13B           | 44 →  0              | 28 → +9              | –36.4 → –15.9 |
> | Pixtral-Large-124B      | 68 → +4              | 56 → +15             | –17.6 →  –1.3 |
> | Pixtral-12B             | 41 → +4              | 36 → +12             | –12.2 → +6.6 |
> | Llama-3.2-Vision-11B    | 36 → +6              | 31 → +9              | –13.9 →  –4.7 |
>
> We find significant recovery of the performance gap for Pixtral-12B, Pixtral-Large-124B, Llama-3.2-Vision-11B, and LLaVA-1.5-13B. We therefore suggest reasoning as a potential avenue to mitigate factual-recall degradation in VLMs without needing excessive pre-training datasets or the resources for native training.
>
> ---
>
> We hope these clarifications and additional results resolve all outstanding concerns and demonstrate the robustness of our findings. We are happy to answer any remaining questions. Thank you for your thoughtful review.

---

> > ### Comment · Reviewer_nkVQ · 2025-08-03
> > **Response to the author**
> >
> > Thanks for the detailed response. I have read it carefully, along with the other reviews, and most of my concerns have been resolved. All things considered, I would like to keep my scores.

---

### Official Review · Reviewer_wjJF · 2025-07-05

**Clarity:** 3
**Significance:** 3
**Originality:** 2
**Rating:** 4
**Confidence:** 3

**Summary:**

This paper investigates the poor performance of multi-modal large models in factual recall tasks. First, the authors propose that the distributed representations of visual information across visual tokens in early layers bypass the factual recall mechanism residing in the early-layer MLPs of the LM backbone, which is the root cause of this phenomenon. Based on this analysis, they then patch early-layer MLP outputs from the LM backbone into the corresponding VLM layers, significantly improving factual recall performance. Finally, the authors introduce a benchmark to systematically evaluate factual recall accuracy and knowledge hallucination in multi-modal settings.

**Questions:**

1. I'm intrigued by the inherent contradiction in VLMs: During pretraining, the primary objective is vision-language alignment, yet empirical evidence shows persistent token space misalignment. Would scaling up pretraining (e.g., with more data/compute) effectively mitigate this issue?
2. Does the proposed activation patching technique impact inference speed? Quantitative metrics are required to evaluate its computational overhead.

**Ethical Concerns:**

["NO or VERY MINOR ethics concerns only"]

**Final Justification:**

Thank you for answering my question; I decided to keep my original score.

**Limitations:**

yes

**Paper Formatting Concerns:**

No concern

**Quality:**

3

**Strengths And Weaknesses:**

**Strengths:**

1. The authors hypothesize that adapter misalignment is the root cause of factual recall degradation in VLMs, and gives the illustration of the Two-Hop problem.
2. The manuscript is explicit and well-organized, and provides visualizations.
3. Experimental validation is sufficient and solid. The proposed method achieves remarkably high accuracy, demonstrating impressive performance compared to baselines, as shown in Figure 5.

**Weaknesses:**

1. While the analysis in Chapter 2 provides illustrative insights, its exclusive reliance on LLaVA-series models raises concerns about generalizability. The observed phenomena may not necessarily hold for other architectural paradigms, requiring validation across more diverse model families.
2. The authors conducted experiments only on the LLaVA-series architectures. Including evaluations on Qwen-VL and InternVL would further strengthen the confidence of the findings.
3. The citation format in the paper has some minor issues. please check it.

---

> ### Author Rebuttal · Authors · 2025-07-31
>
> We sincerely appreciate wjJF’s thoughtful comments and suggestions.
>
> We address all questions and concerns raised, specifically:
> 1. Generalization of our results
> 2. Influence of scaled-up pre-training and alternative architectures on representation and token space
> 3. Inference speed of the patching experiment
>
> We also thank wjJF for careful proofreading and for pointing out minor citation-format issues; we have double-checked all citations and fixed the problems.
>
> ## 1. Generalization of our results
> We appreciate the suggestion to investigate the generalization of our findings by including more diverse models.
>
> To mitigate concerns about the generality of our results, we have:
> 1. Updated the benchmark dataset to ensure a more diverse, realistic, and factually grounded distribution.
> 2. Run the benchmark across a wide range of models, including adapter-based models (LLaVA, Pixtral, Qwen-VL), native models (Llama-4, Gemma 3, Gemini 2.0), and a cross-attention model (Llama 3.2-Vision) covering model sizes from 7B to 124B.
>
> Given inconsistencies in model classifications in the literature, we adopt the following definitions:
> - **Adapter-based**: models that train on top of a pretrained LLM and a pretrained ViT.
> - **Native**: models that train the autoregressive decoder on textual and visual data from the beginning (no separate pretrained LLM).
> - **Cross-attention**: models that use cross-attention layers to incorporate visual information (“Flamingo style”).
>
> ### Results
>
> | **VLM Model**            | **LLM Model**      | **Architecture** | **LLM (%)** | **VLM (%)** | **Δ VLM → LLM (%)** |
> |--------------------------|--------------------|------------------|-----------|-----------|---------|
> | LLaVA-MORE-8B            | Llama-3.1-8B-it    | Adapter          | 41 | 23 | **–43.9** |
> | LLaVA-NEXT-8B            | Llama-3-8B-it      | Adapter          | 41 | 24 | **–41.5** |
> | LLaVA-1.5-7B             | Llama-2-7B-chat    | Adapter          | 30 | 19 | **–36.7** |
> | LLaVA-1.5-13B            | Llama-2-13B-chat   | Adapter          | 44 | 28 | **–36.4** |
> | Pixtral-Large-124B       | Mistral-Large-2    | Adapter          | 68 | 56 | **–17.6** |
> | Pixtral-12B              | Mistral-NeMo       | Adapter          | 41 | 36 | **–12.2** |
> | Qwen-2.5-VL-7B-it        | Qwen-2.5-7B-it     | Adapter          | 30 | 28 | **–6.7** |
> | Qwen-2.5-VL-72B-it       | Qwen-2.5-72B-it    | Adapter          | 49 | 53 | **+8.2** |
> | Llama-4-Maverick         | —                  | Native           | 75 | 71 | **–5.3** |
> | GPT-4o                   | —                  | Native           | 77 | 73 | **–5.2** |
> | Gemini-2.0-Flash         | —                  | Native           | 66 | 63 | **–4.5** |
> | Gemma-3-27B-it           | —                  | Native           | 46 | 46 | **0.0** |
> | Llama-3.2-Vision-11B     | Llama-3.1-8B-it    | Cross-attention  | 36 | 31 | **–13.9** |
>
> ### Findings
>
> - Factual-recall degradation appears in **11 of 13 models tested**.
> - The problem **extends to very large adapter-based models** (Pixtral-Large-124B, –17.6 %) and to the cross-attention model (–13.9 %).
> - Native models exhibit smaller drops (≈ 4–5 %), and Qwen-2.5-VL actually outperforms its backbone (+8.2 %), likely due to its >4T-token multimodal alignment training corpus.
>
> ### Summary
>
> 1. Factual-recall degradation is prevalent (11 / 13 models).
> 2. It affects large adapter-based and cross-attention models.
> 3. Native models, or adapter-based models trained on massive multimodal corpora, show less degradation.
>
> Thus, unless one can afford native end-to-end training or huge multimodal datasets, understanding and mitigating factual-recall degradation is a critical challenge.
>
> ## 2. Pre-training, architecture, and visual representations
>
> We selected Qwen-2.5-VL-7B (adapter-based, >4T alignment training tokens) and Gemma-3-11B (native) to compare with LLaVA-1.5-7B.
>
> We performed:
> 1. Layer-wise linear probing to detect when entity information becomes linearly represented.
> 2. Token-space-alignment experiments measuring cosine similarity between projector outputs and LLM token embeddings.
>
> ### Probing results
>
> | Layer # (32 total) | 0 | 5 | 10 | 15 | 20 | 25 | 30 |
> |--------------------|----|----|----|----|----|----|----|
> | LLaVA-1.5-7B       | **0.18** | 0.40 | 0.81 | 0.96 | 0.98 | 1.00 | 1.00 |
>
> | Layer # (28 total) | 0 | 4 | 8 | 12 | 16 | 20 | 24 |
> |--------------------|----|----|----|----|----|----|----|
> | Qwen-2.5-VL-7B     | **0.78** | 0.91 | 0.96 | 0.99 | 1.00 | 1.00 | 1.00 |
>
> | Layer # (48 total) | 0 | 7 | 14 | 21 | 28 | 35 | 42 |
> |--------------------|----|----|----|----|----|----|----|
> | Gemma-3-11B        | **0.63** | 0.89 | 0.97 | 1.00 | 1.00 | 1.00 | 1.00 |
>
> Both Qwen-2.5-VL-7B and Gemma-3-11B form linear entity representations far earlier than LLaVA-1.5-7B, aligning with their stronger benchmark performance.
>
> Two mechanisms could account for this:
> 1. Better projection alignment with the LLM token space.
> 2. Adaptation of LLM factual-recall circuits to non-textual representations.
>
> ### Token-space alignment
>
> - **Qwen-2.5-VL-7B**: avg. cosine = 0.0778 ± 0.0023
> - **Gemma-3-11B**: avg. cosine = 0.1283 ± 0.0043
>
> Gemma-3’s higher cosine stems largely from tokens aligning with the beginning-of-sentence embedding (≈ 0.33) and therefore carrying no visual content. Thus, early representational availability, not projection/token-space overlap, likely yields the improvement, supporting hypothesis 2. We think carrying out a detailed analysis of how visual representations integrate with LLM mechanisms like factual recall is an exciting avenue for future research.
>
> ## 3. Computational overhead of patching
>
> Our patching round requires:
> - One forward pass through the LLM (text input) to cache MLP activations.
> - One forward pass through the VLM (image + text) to cache MLP activations.
> - One backward pass through the VLM to obtain KL-divergence gradients.
> - A final matrix multiplication of gradient × activation differences.
>
> Total overhead: two forward passes, one backward pass, plus a single matrix multiply. On a single A100, running 100 examples requires ≈ 40 minutes. We emphasize that the method is intended as an experimental setup to support our hypothesis about the role of early-layer MLPs, and not as a practical approach to reduce factual-recall degradation.
>
> ## Additional: reasoning / chain-of-thought (CoT)
> While our patching experiment shows that factual-recall degradation can be reduced by repairing the early-layer MLP outputs, it is not a practical approach. Similarly, it is often not feasible to train a model natively or run large-scale alignment training like Qwen-2.5-VL. Therefore, we ran an experiment where we include a chain-of-thought prompt, allowing the model to reason about the image content and the question for a couple of sentences.
>
> | **VLM Model**           | **LLM (Base → CoT)** | **VLM (Base → CoT)** | **Δ Gap (%)** |
> |-------------------------|---------------------|---------------------|--------------|
> | LLaVA-MORE-8B           | 41 → +5             | 23 → +5             | –43.9 → –39.1 |
> | LLaVA-NEXT-8B           | 41 → +4             | 24 → +6             | –41.5 → –33.3 |
> | LLaVA-1.5-7B            | 30 → +5             | 19 → +0             | –36.7 → –45.7 |
> | LLaVA-1.5-13B           | 44 → +0             | 28 → +9             | –36.4 → –15.9 |
> | Pixtral-Large-124B      | 68 → +4             | 56 → +15            | –17.6 →  –1.3 |
> | Pixtral-12B             | 41 → +4             | 36 → +12            | –12.2 → +6.6 |
> | Llama-3.2-Vision-11B    | 36 → +6             | 31 → +9             | –13.9 →  –4.7 |
>
> We find significant recovery of the performance gap for Pixtral-12B, Pixtral-Large-124B, Llama-3.2-Vision-11B, and LLaVA-1.5-13B. We therefore suggest reasoning as a potential avenue to mitigate factual-recall degradation in VLMs, without needing excessive pre-training datasets or the resources for native training.
>
> ---
>
> We hope these clarifications and additional results resolve all questions and outstanding concerns and demonstrate the robustness of our findings. We are happy to answer any remaining questions. Thank you for your thoughtful review.

---

### Note · Authors · 2025-08-16

We thank the reviewers for their constructive engagement, and the ACs and SACs for their time and consideration. We briefly recap the discussion phase below.

Several strengths were highlighted: a clear mechanistic hypothesis for factual recall degradation in VLMs, strong methodology combining probing and patching, a benchmark useful for isolating modality-specific recall, and causal evidence from patching experiments.

The main concerns centered on generalization beyond LLaVA, alternative explanations, model scale, benchmark diversity, and patching details. We addressed each directly:
- Generalization: Evaluation expanded to 13 models across adapter-based (LLaVA, Pixtral, Qwen-VL), native (Gemma-3, GPT-4o, Llama-4, Gemini-2.0), and cross-attention (LLaMA-3.2-Vision, a Flamingo-style architecture).
- Scale: Very large models like Pixtral-Large-124B still show a –17.6% drop, showing that size alone does not solve the issue.
- Alternative explanations: Added random patching (4%) and back-patching (6%) baselines vs. our early-MLP patching (36%), which isolates early MLPs as the bottleneck.
- Mechanistic evidence: Probing shows that models with minimal degradation (Gemma-3, Qwen-2.5-VL) form early linear entity representations, directly validating our hypothesis.
- Benchmark & heuristics: Replaced ImageNet with WIT for factual diversity; audited slot-selection and exclusions, finding no systematic bias.

We appreciate the two reviewers who explicitly stated their concerns were resolved, with one raising their score following our rebuttal and added results. All new findings will be incorporated into the final version.

We would like to note that reviewer zisd’s final comment, posted close to the end of the discussion period, repeated points already addressed and mischaracterized both our reasoning and patching experiments. They claimed we had not tested fundamentally different architectures despite our inclusion of the exact Flamingo-style model they suggested, plus native and large-scale multimodal-trained architectures. Their request for mechanistic evidence that reasoning reduces “timing misalignment” overlooked our explicit explanation that reasoning produces entity tokens, which directly enables the established LLM recall pathway. No specific follow-up experiment was suggested, leaving no opportunity for further clarification.

We again thank all reviewers for their input, which has substantially strengthened the paper.

---

### Decision · Program_Chairs · 2025-09-17

**Decision:**

Accept (poster)

**Comment:**

This paper deals with multomodal knowledge retrieval and investigates poor performance of LLaVa-style VLMs in factual recall tasks.  The paper attributes this to the VLM architecture in whicb visual information is processed too late.  To counter this the paper treats this as a "two-hop" problem and shows that visual information takes a clean linear form in mid-to-late layers and injecting MLP activations from unimodal model into the VLM can help.

The paper received four positive reviews, albeit all borderline.  Authors are encouraged to integrate feedback from the reviews and rebuttal/discussion phase into the subsequent version of the paper.  The code and benchmark should be open-sourced as it seems it's not explicitly stated.